# Investigation of Spatio–Temporal Changes in Land Use and Heat Stress Indices over Jaipur City Using Geospatial Techniques

**Suresh Chandra [1], Swatantra Kumar Dubey [2], Devesh Sharma [3],\*, Bijon Kumer Mitra [4] and Rajarshi Dasgupta [4]**

[1] Centre of Excellence for Climate Change & Vector-Borne Diseases, ICMR-National Institute of Malaria Research, Department of Health Research, Government of India, New Delhi 110077, India; sureshchandra1987@hotmail.com

[2] Department of Environmental Engineering, Seoul National University of Science & Technology (SeoulTech), Gongneung-ro, Nowon-gu, Seoul 01811, Korea; swatantratech1@gmail.com

[3] Department of Atmospheric Science, School of Earth Sciences, Central University of Rajasthan, Ajmer 305817, India

[4] Integrated Sustainability Center, Institute for Global Environmental Strategies (IGES), 2108-11 Kamiyamaguchi, Hayama 240-0115, Kanagawa, Japan; b-mitra@iges.or.jp (B.K.M.); dasgupta@iges.or.jp (R.D.)

\* Correspondence: deveshsharma@curaj.ac.in

**Abstract:** Heat waves are expected to intensify around the globe in the future, with a potential increase in heat stress and heat-induced mortality in the absence of adaptation measures. India has high current exposure to heat waves, and with limited adaptive capacity, impacts of increased heat waves might be quite severe. This paper presents a comparative analysis of urban heat stress/heatwaves by combining temperature and vapour pressure through two heat stress indices, i.e., Wet Bulb Globe Temperature (WBGT) and humidex index. For the years 1970–2000 (historical) and 2041–2060 (future), these two indicators were estimated in Jaipur. Another goal of this research is to better understand Jaipur land use changes and urban growth. For the land use study, Landsat 5 TM and Landsat 8 OLI satellite data from the years 1993, 2010, and 2015 were examined. During the research period, urban settlement increased and the majority of open land is converted to urban settlements. In the coming term, all months except three, namely July to September, have seen an increase in the WBGT index values; however, these months are classified as dangerous. Humidex's historical value has been 21.4, but in RCP4.5 and RCP8.5 scenarios, it will rise to 25.5 and 27.3, respectively, and slip into the danger and extreme danger categories. The NDVI and SAVI indices are also used to assess the city's condition during various periods of heat stress. The findings suggest that people's discomfort levels will rise in the future, making it difficult for them to work outside and engage in their usual activities.

**Keywords:** heat stress; WBGT index; climate change; land use; humidex index

## 1. Introduction

The occurrence of more extreme climate events has been becoming more frequent and severe as global warming, and causes a distressing effect on human lives [1]. These changes can have both positive and negative impacts on urbanization and human health. Climate change will have a significant impact on metropolitan areas, and it may result in chronic health concerns [2]. Different climate change pathways affect human health between different time periods [3]. India has generated only 2% of total carbon emissions from fossil fuel combustion over the last 100 years [4], which is likely owing to the effects of extreme weather events (NIOO-KNAW, 2017). Human health risks related to climate change can, directly and indirectly, affect older people [5]. An urban heat island (UHI) is a metropolitan area which is significantly warmer than its surrounding rural areas due to

human activities. In metropolitan regions, UHIs tend to amplify the impact of heat waves, and rising temperatures in the area contribute to the likelihood of heat-related deaths [6,7]. In 2003, 3500 deaths were estimated across Europe due to extreme heatwave [8]. People's conditions are worse at night due to the high temperature during heatwaves compared to the high daytime temperature, and it also increases the mortality rate at night-time [9]. The high temperature causes an increase in mortality in metropolitan areas, as well as various health conditions such as heat cramps, weariness, non-fatal heat stroke, and overall discomfort [10]. Climate change models anticipate that a gradual increase in summer temperatures and heat waves will exacerbate the situation [8].

India is most vulnerable to the increased temperature associated with climate change. It is estimated that from 1992, about 25,000 Indian people died because of heat waves [11]. In 2003, heatwaves hit parts of India (Uttar Pradesh, Haryana, Punjab, Rajasthan, Gujarat, Bihar, and Orissa), resulting in a higher fatality rate [12]. As a result of the increased number and frequency of heat waves, the death rate will rise in the future [13]. The climatic approaches such as El Niño-Southern Oscillation (ENSO) and fluctuations in the sea surface temperatures in the Bay of Bengal have been related to the heatwaves over India. Heatwaves may occur as a result of changes in wind direction and a lack of moisture in inland areas, resulting in heat waves. Despite the significant societal impact, no systematic attempt has been made to investigate the primary mechanism of heatwaves in India.

In different parts of the world, some authors employed the WBGT and humidex for heat stress assessments [14,15]. WBGT is an experimental index that was developed by Yaglue and Minard in 1957 and published as an ISO 7243 standard in 1989. It is used in both indoor and outdoor environments. It was recommended to eliminate the time-consuming process of calculating the effective temperature index (ET), which was developed from a series of laboratory investigations about 1920 and quickly became the standard approach for assessing heat stress [16]. Temperature, humidity, radiation, and wind were merged into a single figure that could be utilized for assessment (ISO, 1989). The natural wet bulb temperature, globe temperature, and air temperature are the key determinants of WBGT. The WBGT index's most important strength is its sensitivity to radiant heat and air movement, which are two important factors in estimating the ambient air temperature [15,17]. In tropical and subtropical areas of the world, climate change has resulted in temporal and spatial changes in workplace heat exposure, resulting in occupational health issues. In this regard, the results of prior studies show that WBGT values have been rising in recent years. Wet bulb globe temperature (WBGT) is used as a heat stress indicator for assessment of thermal comfort in environments [15,18,19]. Ref. [20] examined WBGT in the Coimbra region of Portugal and found a strong association between globe temperature of 2.8 percent and natural wet bulb temperature of 2.6 percent and WBGT. Ref. [21] assessed the thermal comfort in 15 regions with the help of WBGT by evaluating the past and future threshold exceedance rates concerning moderate (28 °C), high (32 °C) and extreme (35 °C) temperatures. They are using the WBGT for the 2020s and 2050s with A1B scenarios and in the HadCM3 model, and observed that heat events might become aggravated in regions of tropical humidity and mid-latitude even though the temperature there would be less than the global average, but the absolute humidity is on the rise. The authors of [22] projected the future heat waves in India using the WBGT index using the CMIP5 scenarios data. They used the three representative concentration pathways (RCPs) RCP2.6, RCP4.5, and RCP8.5 for the historical and future period and projected the severe heatwave in the future period. The study aims to calculate the heat stress in the study area and its effects on human health in the past and future scenarios. The state-of-the-art of the research in the study is presented in Section 2. Section 3 data and methodology describes the SWBGT, humidex, and NDVI procedures and defines the simulation flow. Section 4: Results and Discussion presents the simulation's results as well as a discussion on them. The study's findings and the most important outcomes are summarized in Section 5 Conclusions.



## 2. Study Area Description

Jaipur is Rajasthan state's capital, India, also called the Pink city, for its characteristics of the buildings' colour. Jaipur has a population of around 3.15 million people (Census of India, 2011). The city is mostly flat and is flanked on three sides by the Aravalli hill ranges: north, northeast, and east. The rest of the city is made up of a combination of barren ground, low to medium height vegetation, and built-up areas like as highways, buildings, and industries [23]. According to the Köppen climate classification, the Jaipur come under the hot semi-arid climate. It is located at an elevation of 431 m above mean sea level and at 26.92° N latitude and 75.82° E longitude. Jaipur covers approximately 1464 km$^2$ (JDA) area and this study cover the 472 km$^2$ area (Figure 1). Jaipur city has mostly as-associated a flat plain and hills encircle it in the northern, northeast, and east directions. The area around Jaipur city experiences three seasons each year: winter from November to February (cold nights with average air temperatures as low as 3 °C), summer from March to June (very hot during the day with maximum air temperatures as high as 48 °C), and monsoon from July to October (with extensive variations in daily average air temperature due to atmospheric conditions) [22]. The rainfall mainly occurs in the July and August months due to the monsoon. According to Chandra et al. 2018, the percentage change of the urban area of the Jaipur city was 13.54 (1993) to 57.32 (2015) and open land has been decreased by 45.84 (1993) to 19.4 (2015) [24]. They also explained the urban city expansion in the north, west, and south direction.

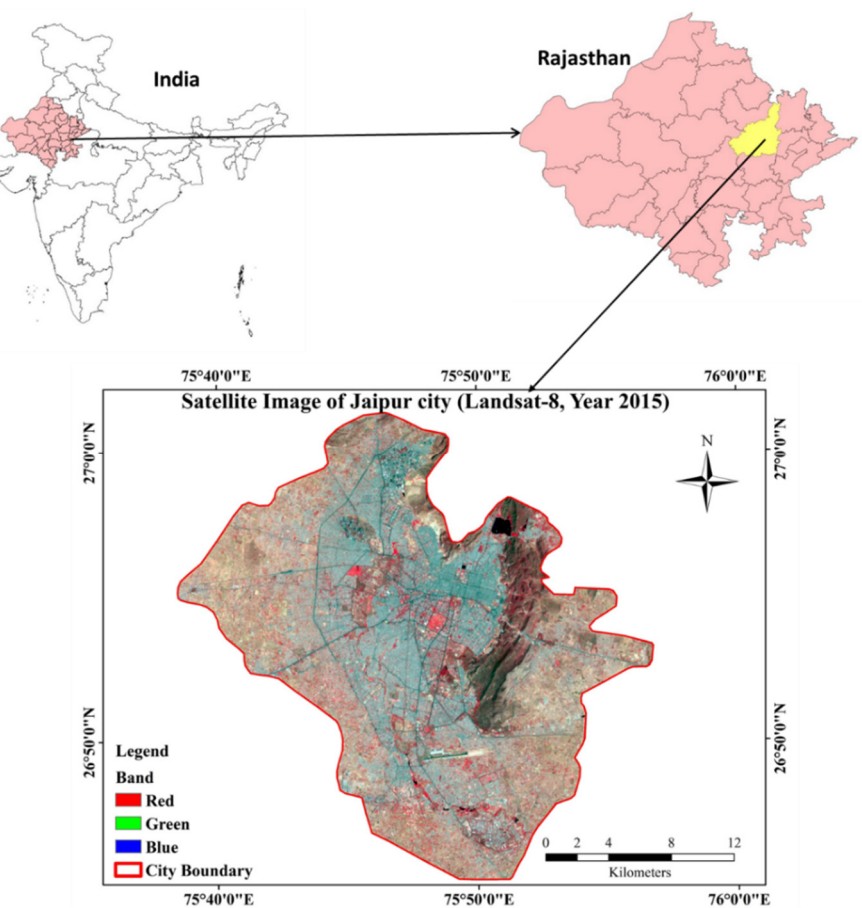

**Figure 1.** Study area map (Jaipur city).

## 3. Materials and Methods

The heat stress indicators were calculated using WorldClim's historical and future datasets. The WorldClim portal (http://worldclim.org (accessed on 2 May 2016) provides

free access to WorldClim datasets for many climate indicators. Long-term average monthly climate data of maximum temperature and vapour pressure were acquired from the World-Clim data portal for the historical period (1970–2000) and future period 2050s (2041–2060) RCP4.5 and RCP8.5 scenarios. Table 1 lists all of the GCMs that were employed in the heat stress analysis. For the past and future eras, this study calculates two heat stress indicators for Jaipur. Monthly ensemble 17 GCMs are used to forecast the research area's future heat stress indices for the future timeframe.

**Table 1.** Detailed information of the GCMs (CMIP5) data of RCP4.5 and RCP8.5.

| GCMs Information | Data Information |
|---|---|
| ACCESS1-0(AC), BCC-CSM1-1(BC), CCSM4(CC), CNRM-CM5(CN), GFDL-CM3(GF), GISS-E2-R(GS), HadGEM2-AO(HD), HadGEM2-CC(HG), HadGEM2-ES(HE), INMCM4(IN), IPSL-CM5A-LR(IP), MIROC-ESM-CHEM(MI), MIROC-ESM(MR), MIROC5(MC), MPI-ESM-LR(MP), MRI-CGCM3(MG), NorESM1-M(NO) | Monthly average maximum temperature (°C*10) GHG Scenarios: RCP4.5; RCP8.5 |

This analysis was conducted by combining temperature and vapour pressure through two heat stress indices, namely Simplified Wet Bulb Globe Temperature (SWBGT) and humidex. Many researchers used the SWBGT indicators to estimate the general heat stress index at various spatial and temporal scales [25,26].

The Australian Bureau of Meteorology [21] suggested the SWBGT indicator for spatial analysis. Equation (1) is used to calculate the *SWBGT* of Jaipur city.

$$SWBGT = 0.567Ta + 0.393e + 3.94 \tag{1}$$

where, *Ta* and *e* represent the air temperature (°C) and water vapour pressure (hPa) near the surface.

The humidex index was developed in Canada to estimate the humidity and consequence of high temperature on human health. The humidex indicator is assessed by using Equation (2) [27]:

$$Humidex = Ta + \left(\frac{5}{9}\right)(e - 10) \tag{2}$$

where, *Ta* is air temperature (°C) and *e* is the water vapour pressure (hPa) near the surface.

After an assessment of these indices, different categories are allocated based on these values. Each group represents a particular kind of condition and is linked with the heat stress situation for their effect on human health. Table 2 provides the classes of heat stress along with their consequence on human health.

**Table 2.** Categories of the heat stress, WBGT and humidex index with human effects.

| Heat Stress Category | WBGT Index | Humidex Index | Inferences |
|---|---|---|---|
| Extreme danger | Greater and equal to 40 | Greater and equal to 46 | Dangerous and the risk of heat stroke |
| Danger | 34–39 | 38–45 | Very uncomfortable and avoid physical exertion |
| Extreme caution | 28–33 | 30–37 | Little uncomfortable |
| Caution | 22–27 | 20–29 | Comfortable |

Source: http://www.crh.noaa.gov, http://www.ec.gc.ca/meteo-weather/ (accessed on 6 August 2016).

### 3.1. Image Classification and Accuracy Assessment

The study area is divided into five key groups using a supervised technique with the maximum likelihood classification method: water body, vegetation, urban settlement, open land, and hilly area/rocky area. The Kappa technique was used to examine the categorization accuracy [28,29].

Kappa coefficient (*k*) for the image classification is as follows:

$$k = \frac{N \sum_{i=1}^{r} xii - \sum_{i=1}^{r} xi + *xi + 1}{N^2 - \sum_{i=1}^{r} xi + *xi + 1} \tag{3}$$

$$k = \frac{(Total\ sum\ of\ correct) - Sum\ of\ the\ all\ the\ (row\ and\ column\ total)}{Total\ squared - Sum\ of\ the\ all\ the\ (row\ and\ column\ total)} \tag{4}$$

The Kappa coefficient should never be greater than or equal to one. The high Kappa value indicates accurate land use class information. According to [30] Monserud and Leemans (1992), Kappa coefficients ranging from 0.55 to 0.7 indicate good agreement, 0.7 to 0.85 indicate very good agreement, and values more than 0.85 indicate excellent agreement between image and ground.

### 3.2. Normalized Difference Vegetation Index (NDVI)

Vegetation cover plays a vital role in diminishing the conservation issues in urban areas. As indicated by Batista et al. 1997, the NDVI esteems went from −1 for the non-vegetated area to +1 for vegetation [31]. For the NDVI estimation red band and visible range band and the NIR band are utilized. The NDVI calculation is as follows:

$$NDVI = \frac{(Band\ 4 - Band\ 3)}{(Band\ 4 + Band\ 3)} \tag{5}$$

### 3.3. Soil-Adjusted Vegetation Index Calculate (SAVI)

The *SAVI* index also plays a role in the vegetation cover, but it adds the area's background soil conditions. *SAVI* calculation is as follow:

$$SAVI = (1 + L) * (band4 - band3)/(band4 + band3 + L) \tag{6}$$

where the TOA reflectance is used for each band and L is a soil brightness correction factor. From Huete (1988), *L* = 0.5 is used in most conditions. Figure 2 shows the methodology and the climatic data used in the study.

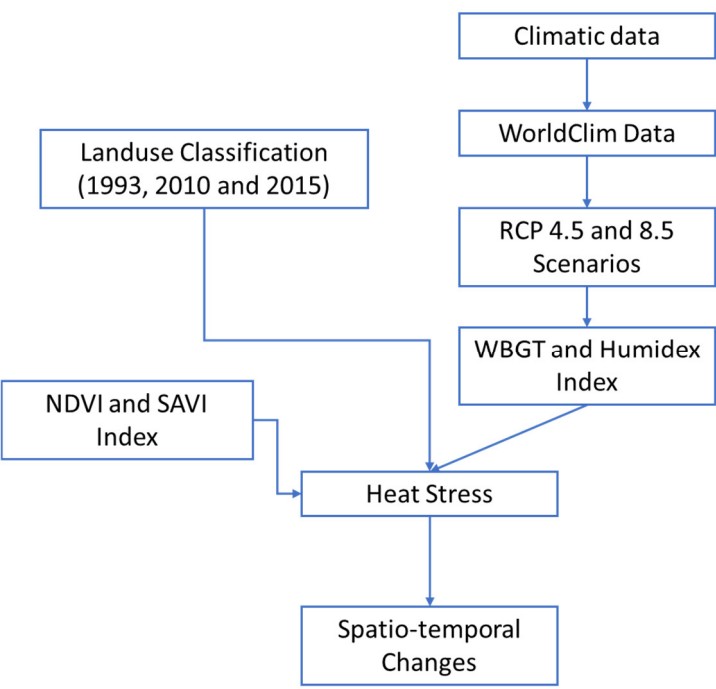

**Figure 2.** Flowchart of methodology and data used.

## 4. Results and Discussions

Heat stress is on the rise in various countries of the world, including India, and is to blame for the rising level of human misery. Heat stress is becoming more severe in cities as a result of urbanization and greenhouse gas emissions.

### 4.1. Land Used Classification

Water body, vegetation, urban settlement, open land, and hilly terrain/rocky area are the five primary land use types evaluated in this study. Land use classifications are carried out for 3 years: 1993, 2010, and 2015. The accuracy of the classified map was determined by a random selection of 330 points for each year. The overall accuracy of the classified maps was found to be 0.92, 0.97, and 0.95 for selected years. According to Table 3, the Kappa coefficients for the indicated years are 0.88, 0.95, and 0.93. In comparison to ground reality, the classified land use accuracy is shown to be good.

**Table 3.** Accuracy assessment of the land cover types.

| | | | Users Accuracy % | | | | |
|---|---|---|---|---|---|---|---|
| **Year** | **Water** | **Vegetation** | **Urban Settlement** | **Open Land** | **Hilly/Rocky Area** | **Overall Accuracy** | **Kappa Coefficient** |
| 1993 | 100.0 | 95.4 | 96.7 | 97.9 | 69.8 | 0.92 | 0.88 |
| 2010 | 100.0 | 94.7 | 100.0 | 96.6 | 91.2 | 0.97 | 0.95 |
| 2015 | 100.0 | 95.7 | 97.2 | 92.7 | 88.6 | 0.95 | 0.93 |
| | | | Producer Accuracy % | | | | |
| **Year** | **Water** | **Vegetation** | **Urban Settlement** | **Open Land** | **Hilly/Rocky Area** | | |
| 1993 | 100.0 | 98.41 | 87.88 | 90.73 | 91.67 | | |
| 2010 | 100.0 | 97.83 | 98.21 | 95.45 | 100.00 | | |
| 2015 | 100.0 | 94.74 | 99.28 | 86.44 | 93.94 | | |

Table 4 shows the total area covered by various categories and their percent coverage. It has been observed that the urban settlement of Jaipur city has grown over time. It was 63.9 km$^2$ in 1993, but by 2015, it expanded to 270.47 km$^2$. This indicates that during the course of 22 years, the area has changed nearly four times. In 2015, over 43.78% change of the studied area was under settlement, compared to the entire area. These trends suggest that the city is rapidly expanding, and it accelerated significantly after 2010.

**Table 4.** Land use area and percent change of different years.

| Class Name | Area 1993 | Area 2010 | Area 2015 | % Change (2010–1993) | % Change (2015–2010) | % Change (2015–1993) |
|---|---|---|---|---|---|---|
| Water. | 0.4 | 0.9 | 0.8 | 0.10 | −0.01 | 0.09 |
| Vegetation | 84.4 | 88.6 | 45.7 | 0.87 | −9.09 | −8.21 |
| Urban Settlement | 63.9 | 166.5 | 270.5 | 21.75 | 22.03 | 43.78 |
| Open Land | 216.3 | 159.8 | 91.5 | −11.96 | −14.47 | −26.44 |
| Hilly/Rocky Area | 106.9 | 56.1 | 63.4 | −10.76 | 1.55 | −9.21 |

Figure 3 depicts the spatial distribution and patterns of land cover change during the three years. The image clearly shows the evolution of urban settlement in Jaipur city. In comparison to the 1993 map, there is a significant rise in of urban area in the Jaipur and found the maximum land use was converted into the urban settlement.

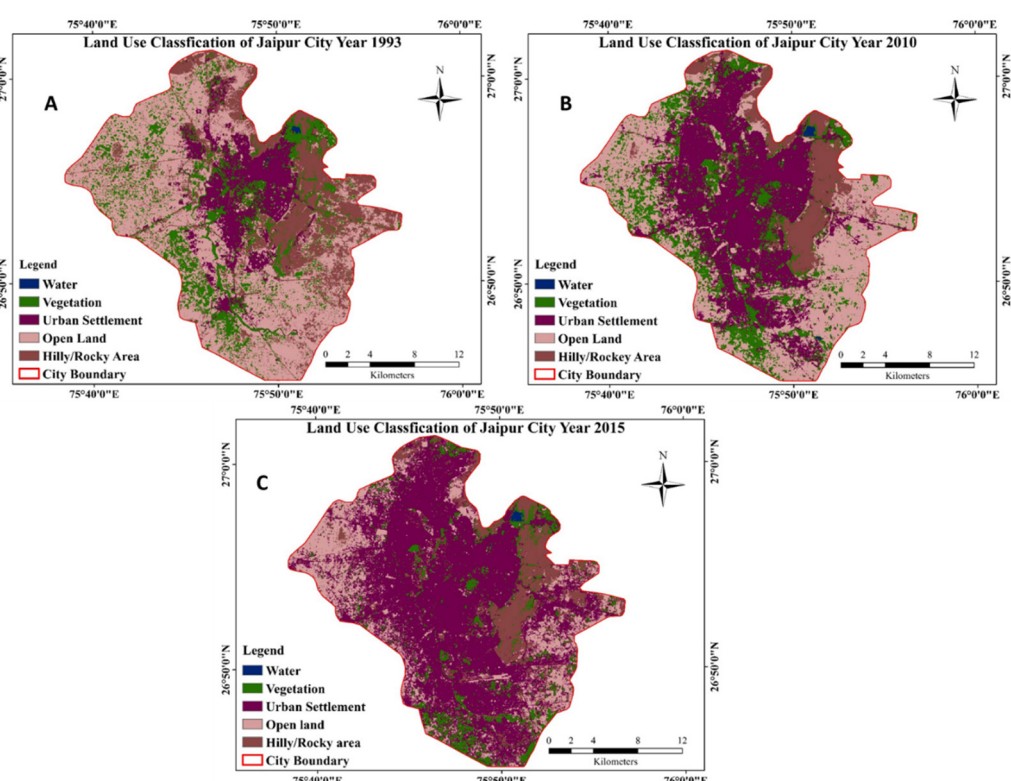

**Figure 3.** LULC maps for different years (**A**) 1993 (**B**) 2010 (**C**) 2015.

### 4.2. Humidex Index

For the historical and future RCP4.5 and RCP8.5 scenarios, the humidex index was calculated on a monthly and seasonal basis. All of the monthly and seasonal data were shown in Table 5. The lowest humidex was recorded in the month of January. The historical minimum humidex value has been 21.4, and in the RCP4.5 and RCP8.5 scenarios, it will

rise to 25.5 and 27.3, respectively. The highest humidex values are observed to be 39.5, 43.2, and 46.4 for the historical and two future RCPs in the May month. In Table 5, the May and June months show the danger conditions in the humidex index for all three cases, but the RCP4.5 and RCP8.5 show the danger and extreme danger conditions in most of the months.

**Table 5.** Average monthly variation in humidex for historical and future periods.

|  | Historical | RCP4.5 | RCP8.5 | Historical | RCP4.5 | RCP8.5 |
|---|---|---|---|---|---|---|
| January | 21.4 | 25.5 | 27.3 | C | C | C |
| February | 24.2 | 28.2 | 30.4 | C | C | EC |
| March | 29.9 | 34.1 | 36.7 | EC | EC | EC |
| April | 35.8 | 39.9 | 42.7 | EC | **D** | **D** |
| May | **39.5** | **43.2** | **46.4** | **D** | D | **ED** |
| June | **39.5** | 41.5 | 44.3 | **D** | D | D |
| July | 39.1 | 36.0 | 38.2 | D | EC | D |
| August | 37.7 | 33.4 | 35.4 | D | EC | EC |
| September | 37.7 | 35.0 | 37.3 | D | EC | D |
| October | 35.3 | 35.6 | 38.3 | EC | EC | D |
| November | 30.2 | 31.5 | 33.7 | EC | EC | EC |
| December | 25.5 | 26.9 | 25.5 | C | C | EC |
| **Winter** | 23.7 | 26.9 | 27.7 | C | C | EC |
| **Monsoon** | **38.5** | 36.5 | 38.8 | **D** | EC | D |
| **Summer** | 35.1 | **39.1** | **41.9** | EC | **D** | D |
| **Autumn** | 32.8 | 33.6 | 36.0 | EC | EC | EC |

C—caution; EC—extreme caution; D—danger; ED—extreme danger.

Figure 4 depicts the spatial distribution of Humidex for all of the months in the past. The months of May and June are classified as Danger and Extreme Danger. From January to May, the Humidex values rise, then begin to decrease until the month of December. There is a slight rise in value in September and October months compared to the decreasing trend. In majority of the months over the historical period, the humidex is high in the southeast and west.

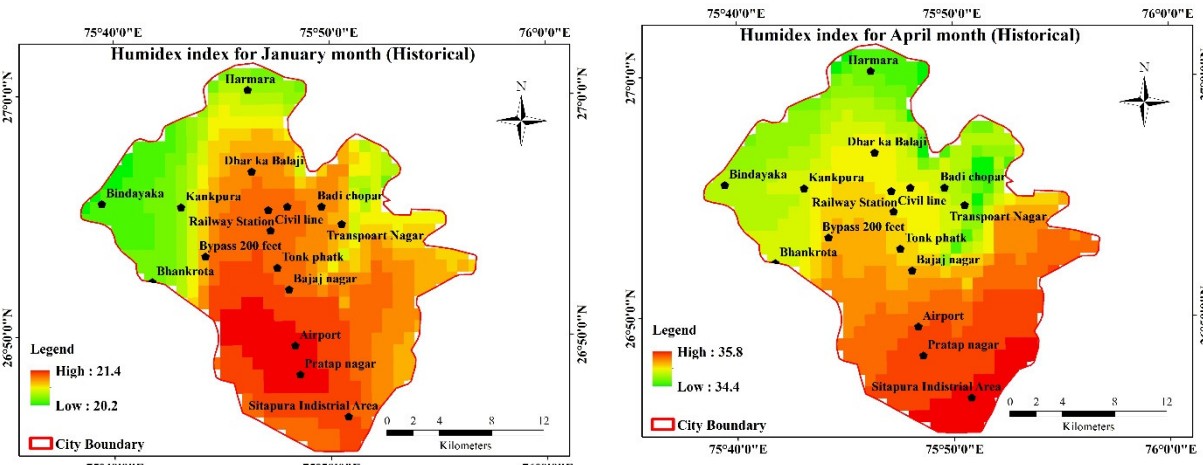

**Figure 4.** *Cont.*

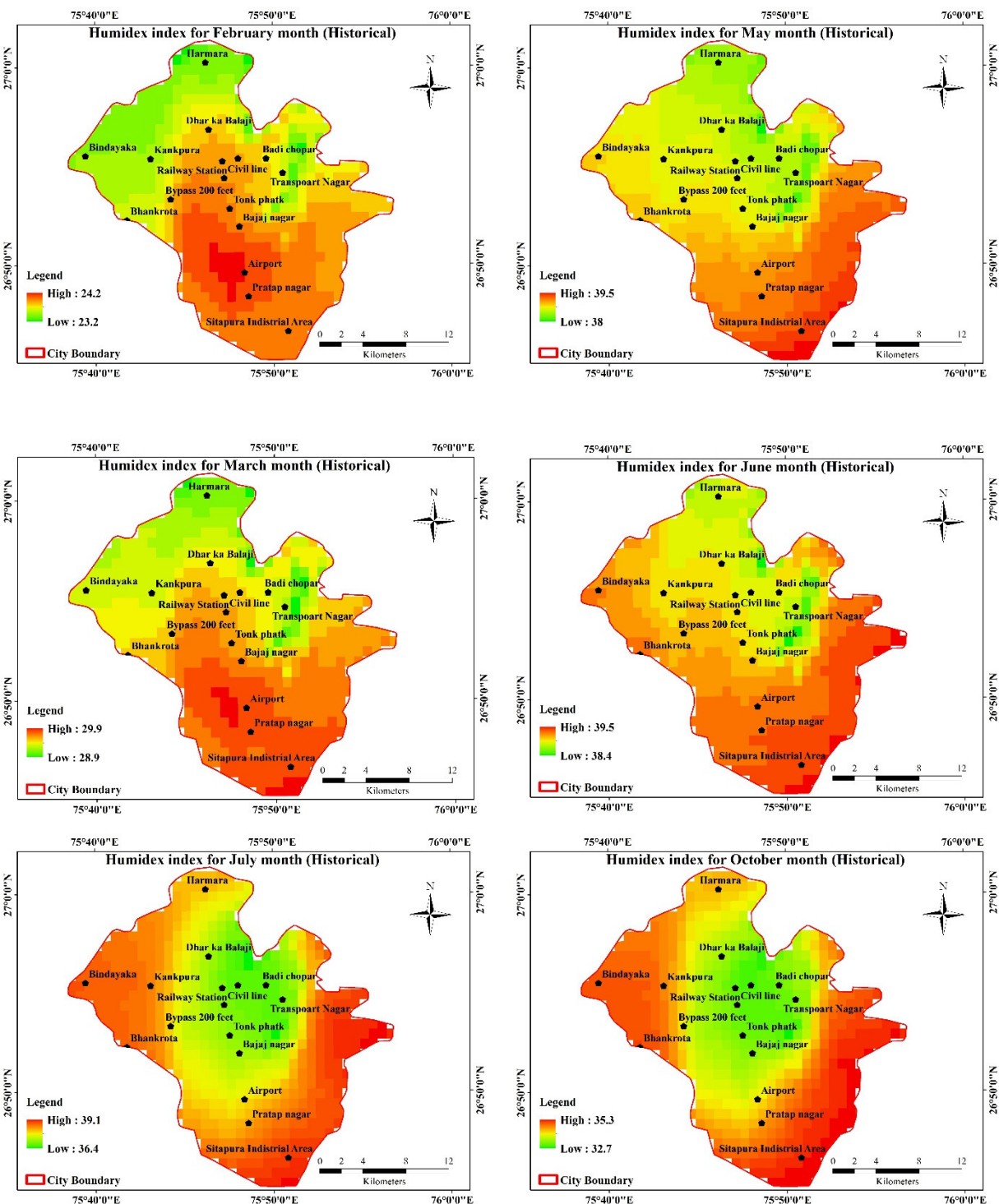

**Figure 4.** *Cont.*

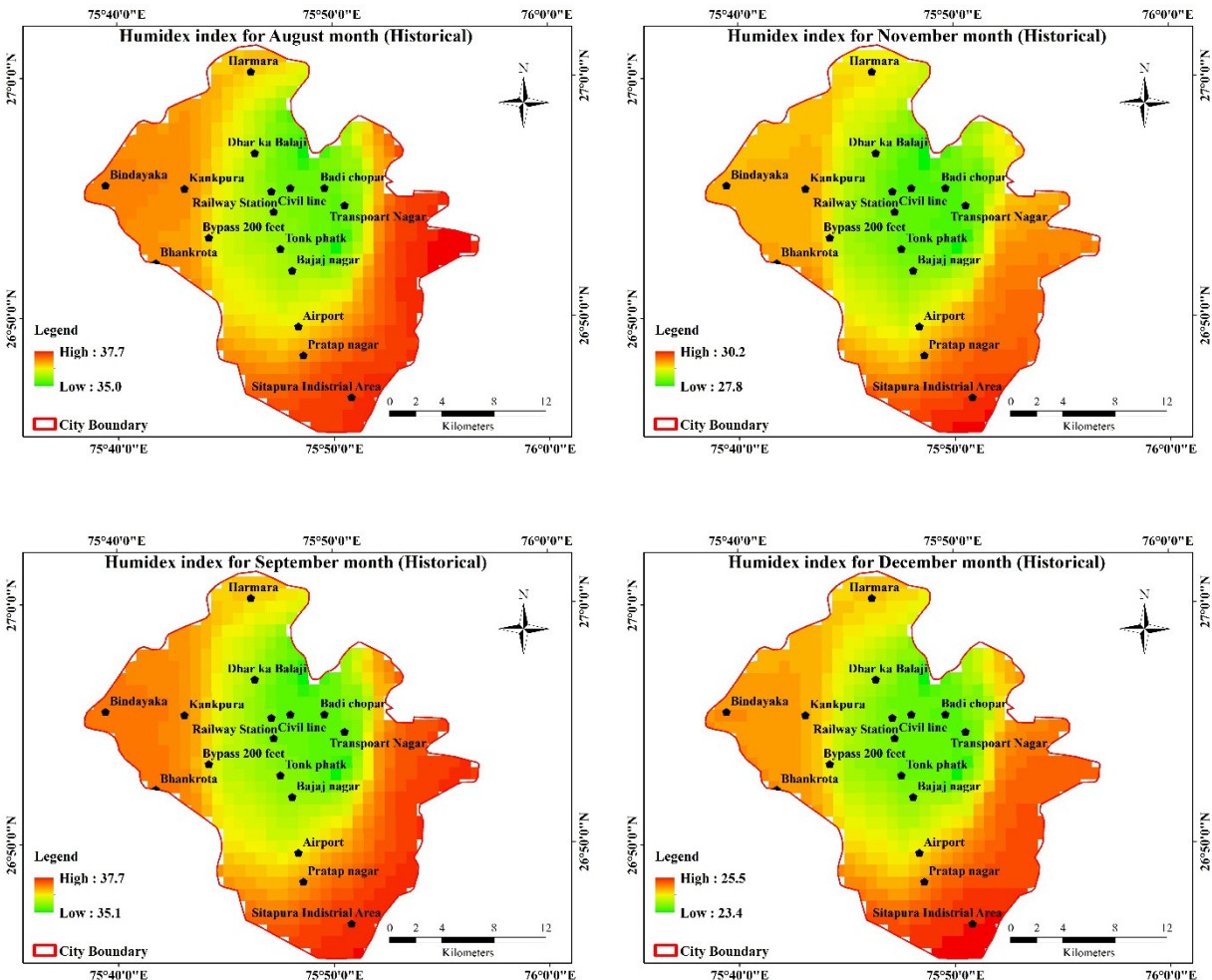

**Figure 4.** Spatial variation of humidex for historical in January to December months.

The spatial maps of humidex variations for January to December for the future RCP4.5 and RCP8.5 scenarios are shown in Figures 5 and 6. The majority of the month in these statistics depicts danger and extreme danger conditions in hypothetical futures. The months of May and June exhibit a danger situation, and the majority of the months fall into the danger and extreme dangerous categories. In the figure, the area with low values is represented by the colour green, while the area with high values is represented by the colour red. The humidex is elevated in the east and south as well as in a small portion of the west side between RCP4.5 and RCP8.5.

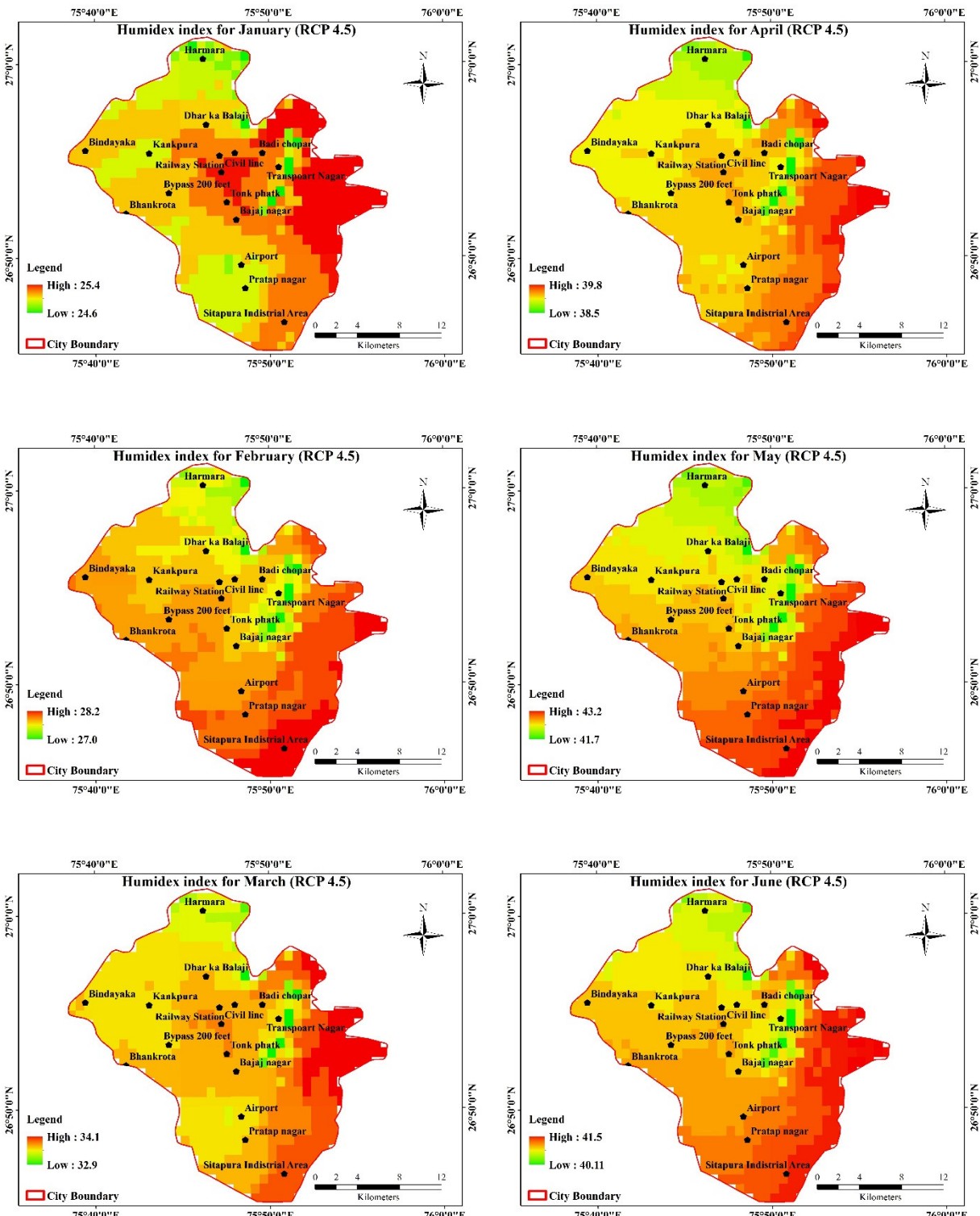

**Figure 5.** *Cont.*

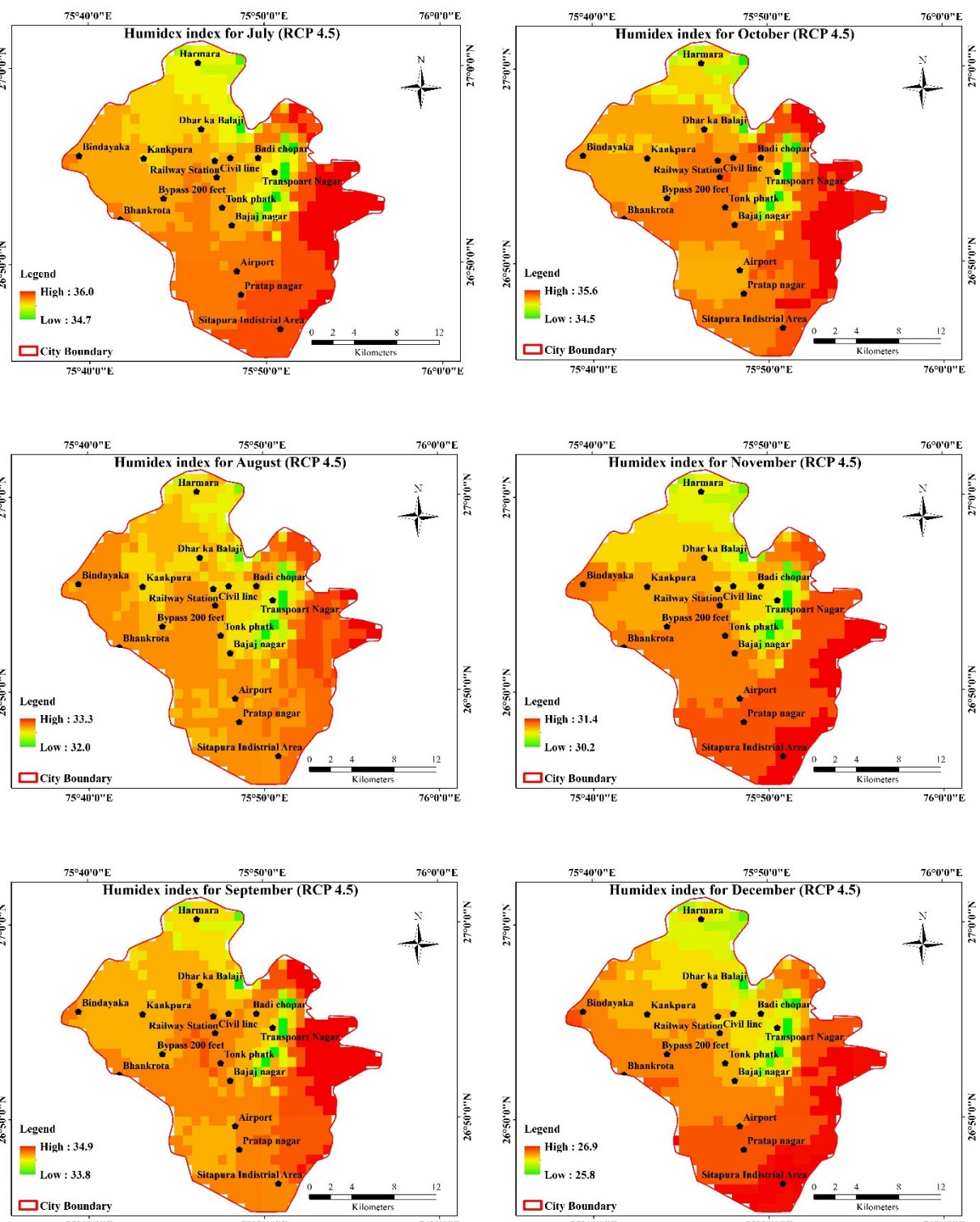

**Figure 5.** Spatial variation of the humidex for future (RCP4.5) in January to December months.

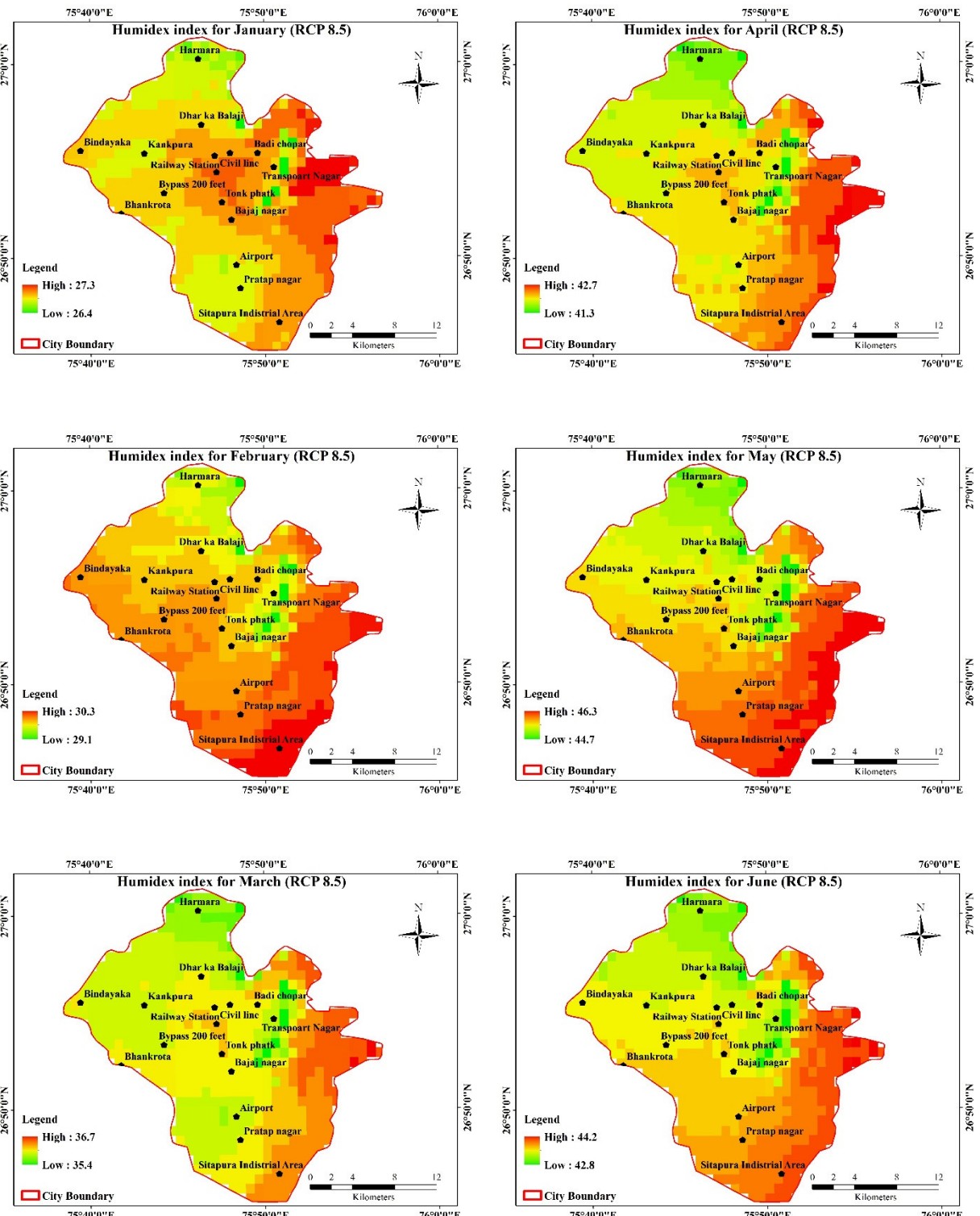

**Figure 6.** *Cont.*

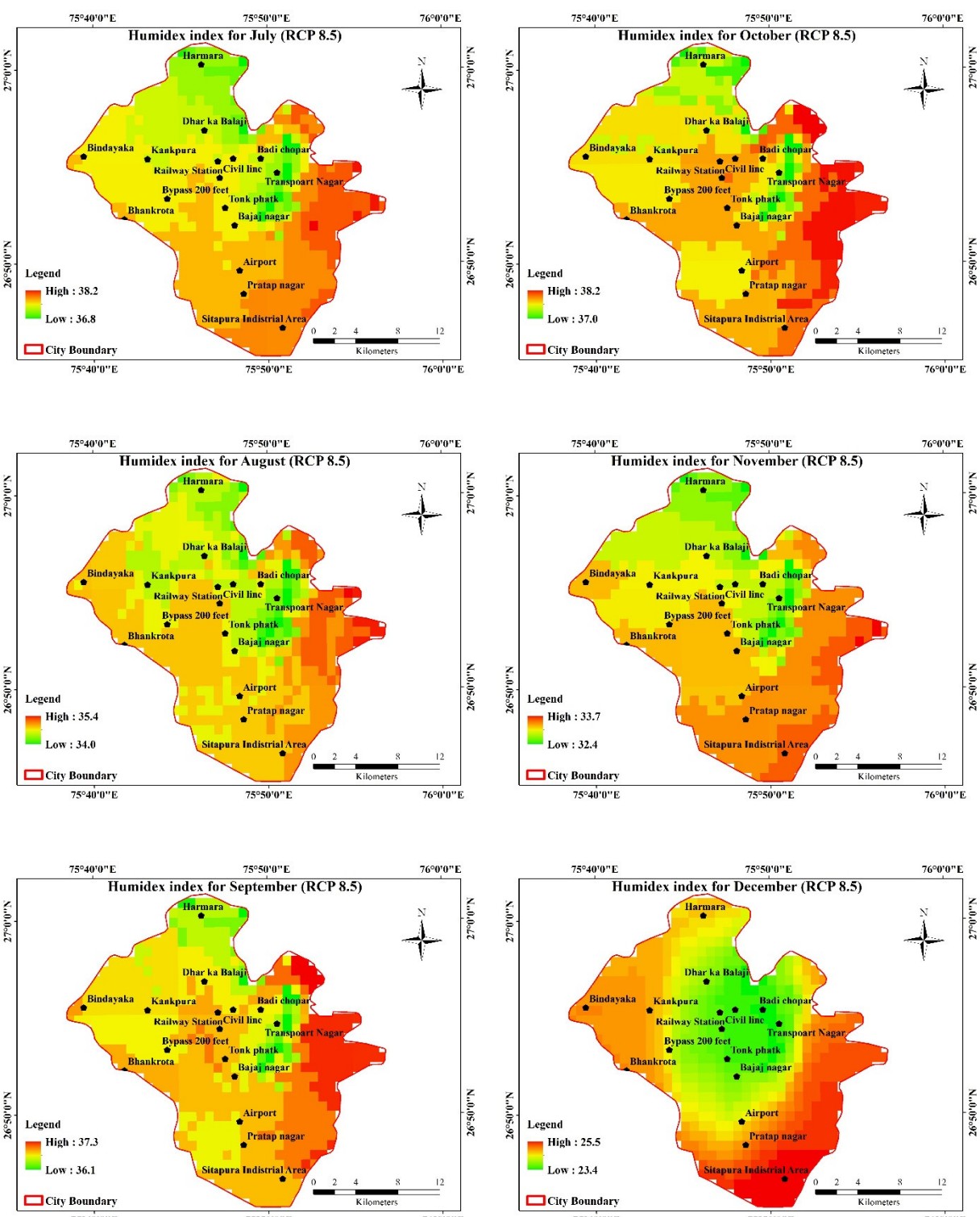

**Figure 6.** Spatial variation of the humidex for future (RCP8.5) in January to December months.

The monthly difference in humidex for historical and projected RCP4.5 and RCP8.5 is shown in Figure 7. In the three months, July to September, as well as throughout the monsoon season, humidex displays a drop. Because RCP8.5 represented the high emission scenario, there is always a significant disparity between RCP8.5 and RCP4.5. It has been demonstrated that the seasonal analysis helps to explain how the severe category shifts.

By the year 2050, the summer season displays a shift from extreme caution to danger and a rise in temperature in the urban region. Figure 8 displays the seasonal humidex variations for the past and future of the city border. The monsoon and autumn seasons show the maximum humidex value in all scenarios and cover the city's east, west, and north direction.

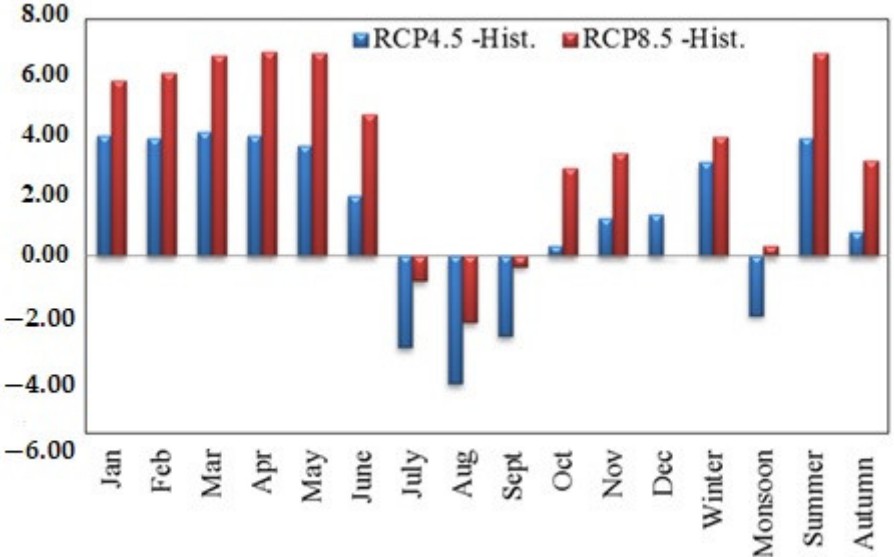

**Figure 7.** Difference in humidex values for future scenarios (2050s).

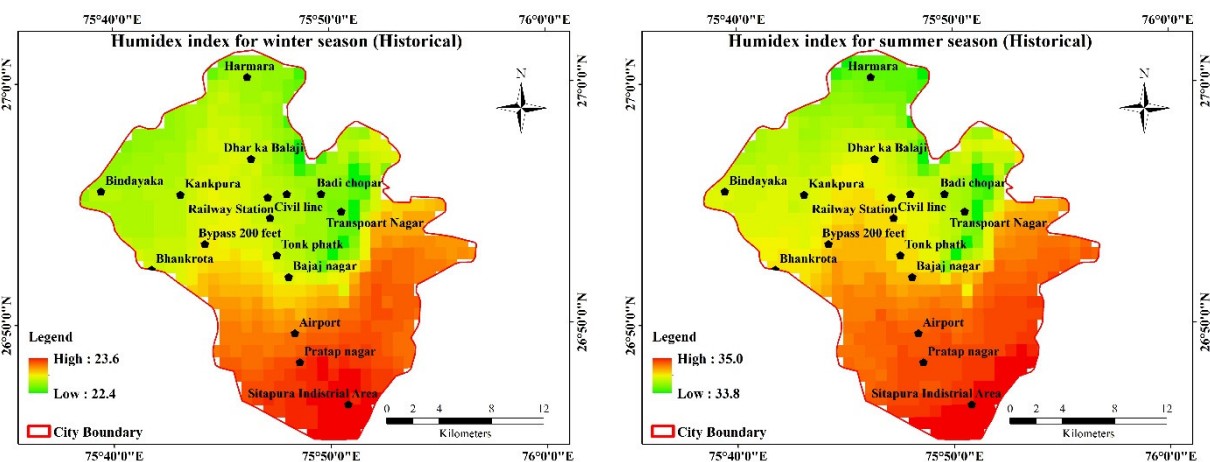

**Figure 8.** *Cont.*

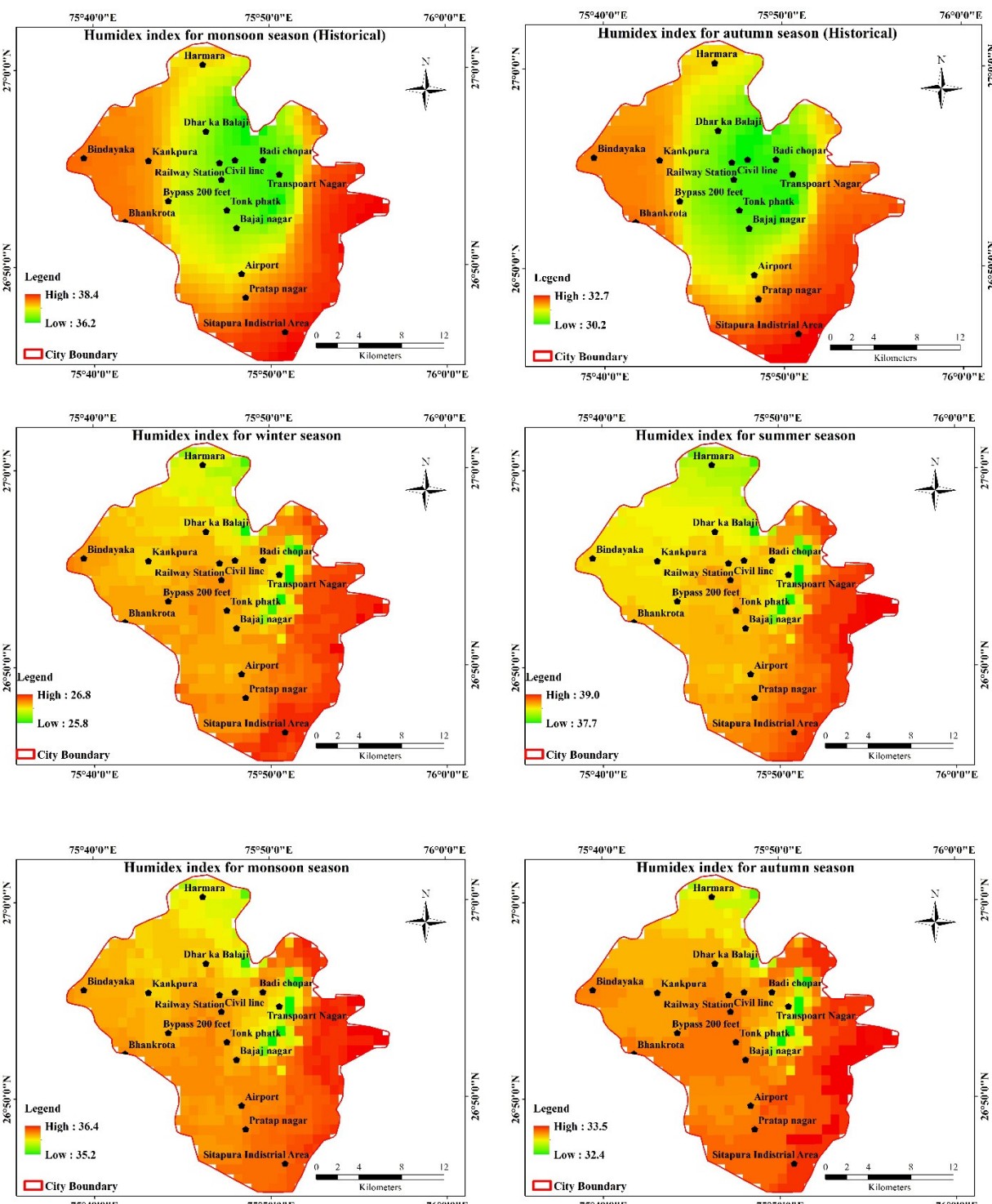

**Figure 8.** *Cont.*

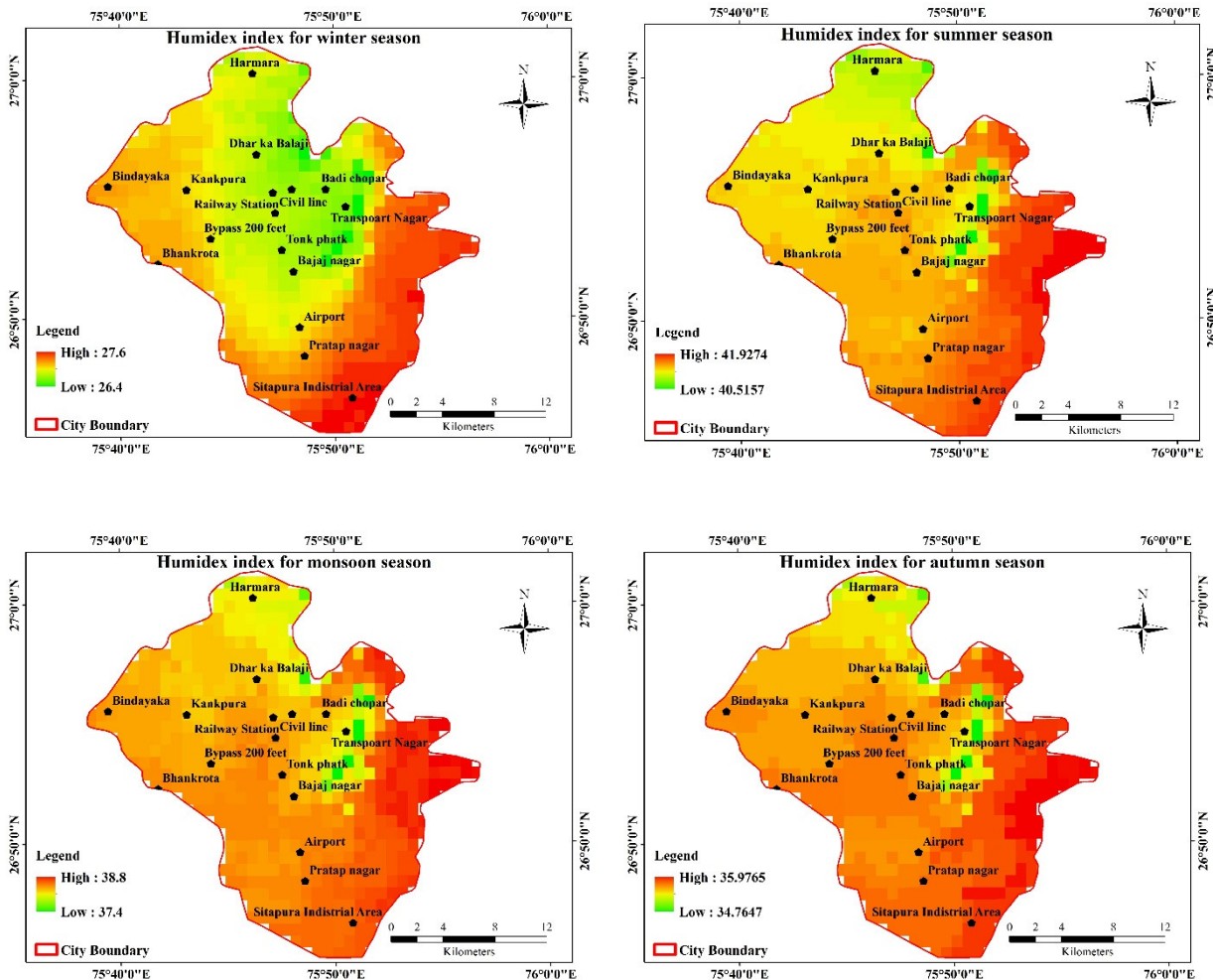

**Figure 8.** Seasonal map of the humidex index of Jaipur city (Historical, RCP4.5, RCP8.5).

*4.3. WBGT Index*

The WBGT index is computed in this study on a monthly and seasonal basis for both past and future periods. The average monthly seasonal fluctuations in the WBGT indicator, together with its stress category, are shown in Table 6. In all three scenarios—historical, RCP4.5, and RCP8.5—the danger categories are visible from June through September. In January, the value is at its lowest, and in June and July, it is at its highest. However, the WBGT, high in the monsoon season of RCP scenarios and the danger situation of the heat of the city of Jaipur, are shown in the season-wise calculation. The correlation coefficient of ESI and environmental parameters of wet temperature, dry temperature, solar radiation, and relative humidity was obtained as 0.88, 0.96, 0.4, and −0.7, respectively, in a study by [32] Hajizadeh et al. (2016), which aimed to investigate the correlation between the environmental stress index (ESI) and WBGT index in a hot and dry climate.

As with the humidex, the lowest values of WBGT are observed for January month in the historical and future periods. The historical value of WBGT is 19.7, which will increase to 22.0 for RCP4.5 and 23.1 for RCP8.5 (Table 6). In the historical period, the high value of WBGT is obtained in the month of July, but it shifts to June for the future period. It is also observed that WBGT values are projected to decrease in the monsoon season (July to September) with the heat stress category of danger. The monthly pattern of values is similar for humidex, increasing from January to June/July and then further decreasing until December. Some cases of a shift from the existing caution condition to extreme caution condition in March, November, and December. The spatial variance of

WBGT in Jaipur city for the past and the future is explained in Figures 9–11. In Figure 9, the southern half of the city showed the greatest changes when compared to other places. Figures 10 and 11, which depict possible futures and determine the city's danger condition, show the same pattern. These show the monthly variation of WBGT values for a future period (both scenarios) compared with historical data. The indicator's value decreases throughout a three-month period from July to September, indicating a decline in indicators during the monsoon season. These areas came under the industrial zones and cover half of the city area. The green colour represents the area with low values, whereas the red colour represents high values.

**Table 6.** Average monthly variation in WBGT for historical and future periods.

|  | Historical | RCP4.5 | RCP8.5 | Historical | RCP4.5 | RCP8.5 |
|---|---|---|---|---|---|---|
| January | 19.7 | 22.0 | 23.1 | C | C | C |
| February | 21.2 | 23.5 | 24.8 | C | C | C |
| March | 24.7 | 27.1 | 28.6 | C | EC | EC |
| April | 28.2 | 30.5 | 32.1 | EC | EC | D |
| May | 30.3 | 32.5 | 34.2 | EC | EC | D |
| June | 34.6 | **35.8** | **37.3** | **D** | **D** | **D** |
| July | **37.3** | 35.5 | 36.8 | **D** | **D** | **D** |
| August | 36.7 | 33.4 | 35.4 | **D** | **D** | **D** |
| September | 34.9 | 33.4 | 34.7 | **D** | **D** | **D** |
| October | 30.0 | 30.2 | 31.7 | EC | EC | EC |
| November | 25.5 | 26.2 | 27.5 | C | C | EC |
| December | 22.2 | 23.0 | 22.1 | C | C | EC |
| **Winter** | 21.0 | 22.8 | 23.3 | C | C | C |
| **Monsoon** | **35.9** | **34.5** | **36.1** | **D** | **D** | **D** |
| **Summer** | 27.7 | 30.0 | 31.6 | EC | EC | EC |
| **Autumn** | 27.8 | 28.2 | 29.6 | EC | EC | EC |

C—caution; EC—extreme caution; D—danger; ED—extreme danger.

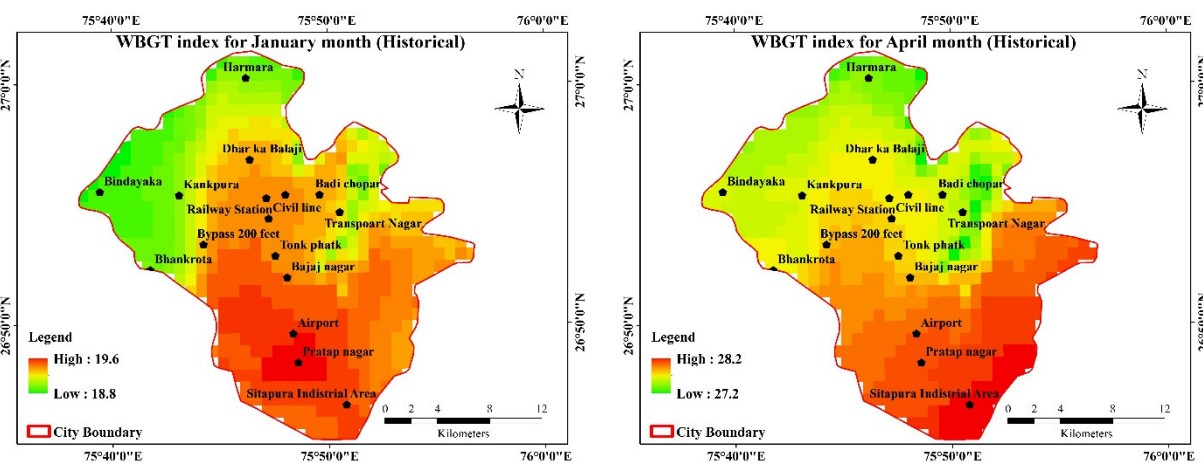

**Figure 9.** *Cont.*

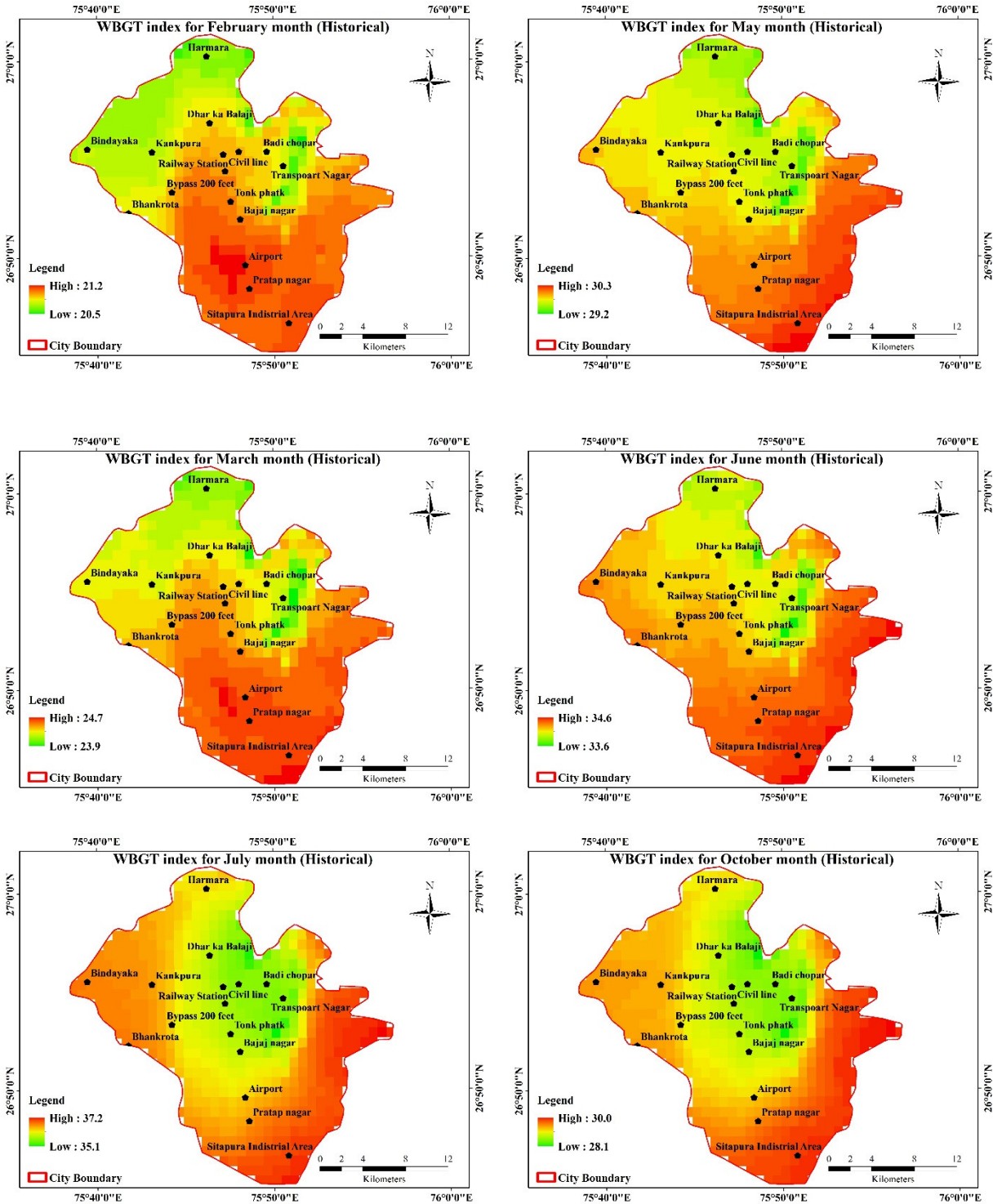

**Figure 9.** *Cont.*

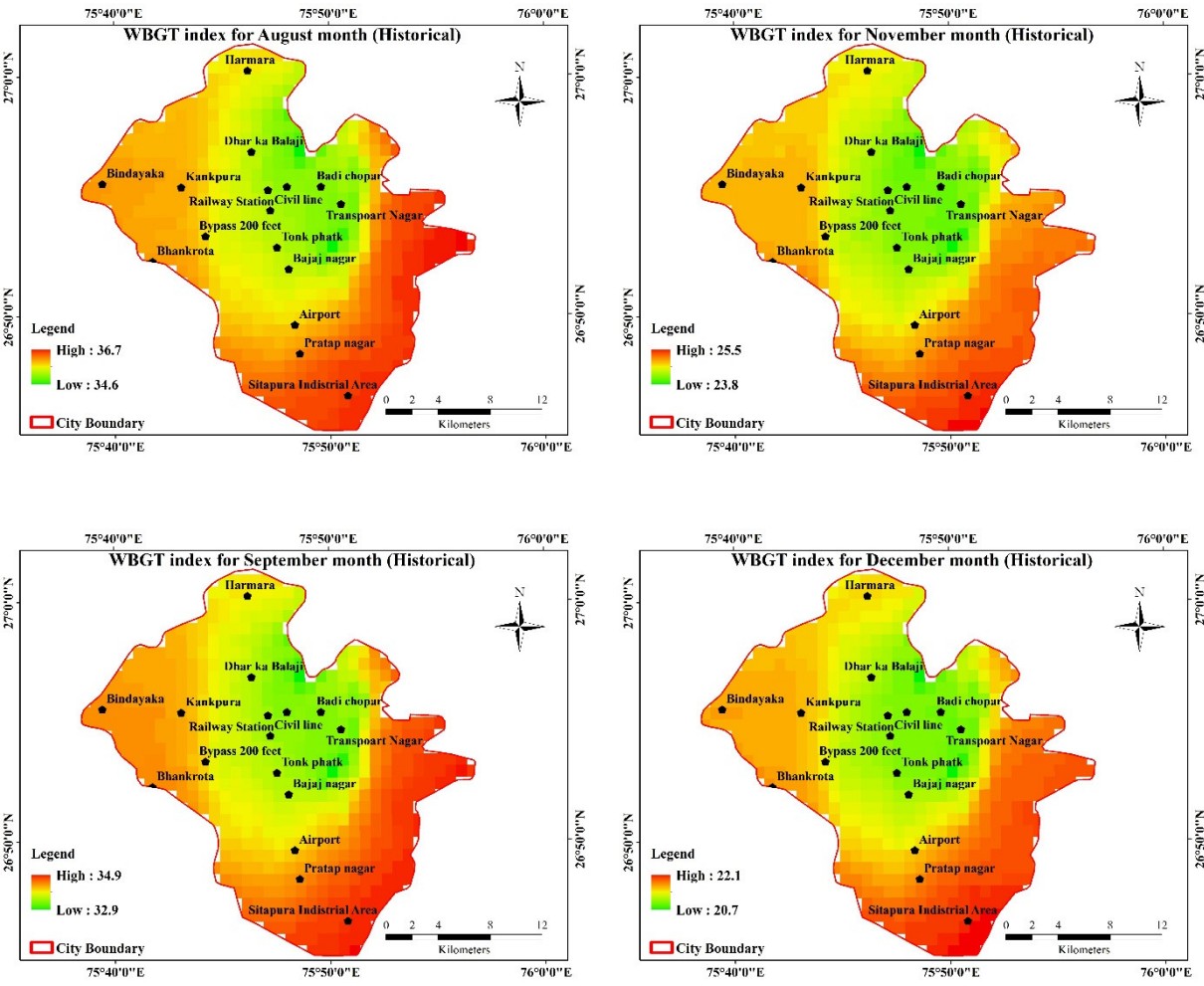

**Figure 9.** Spatial variation of WBGT for the historical period.

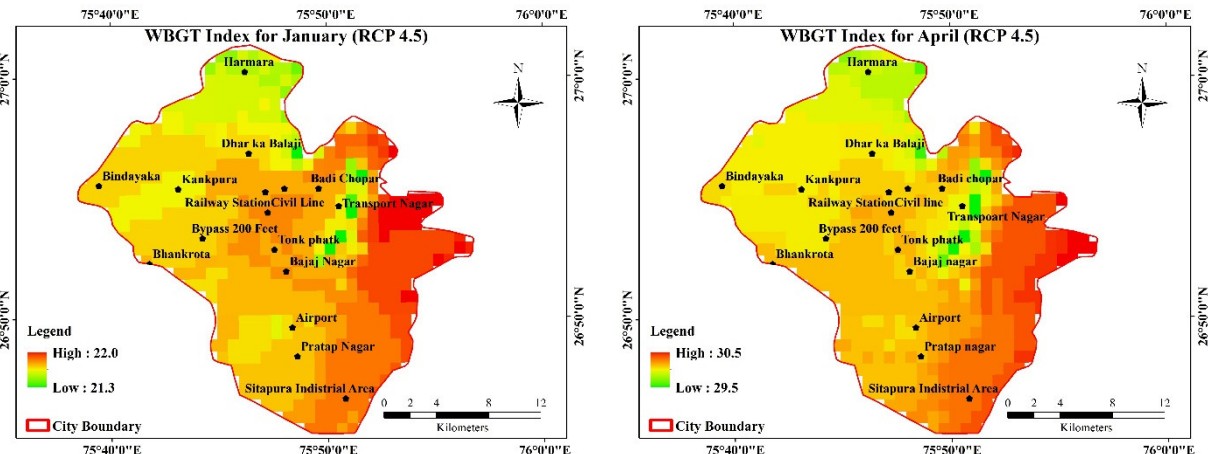

**Figure 10.** *Cont*.

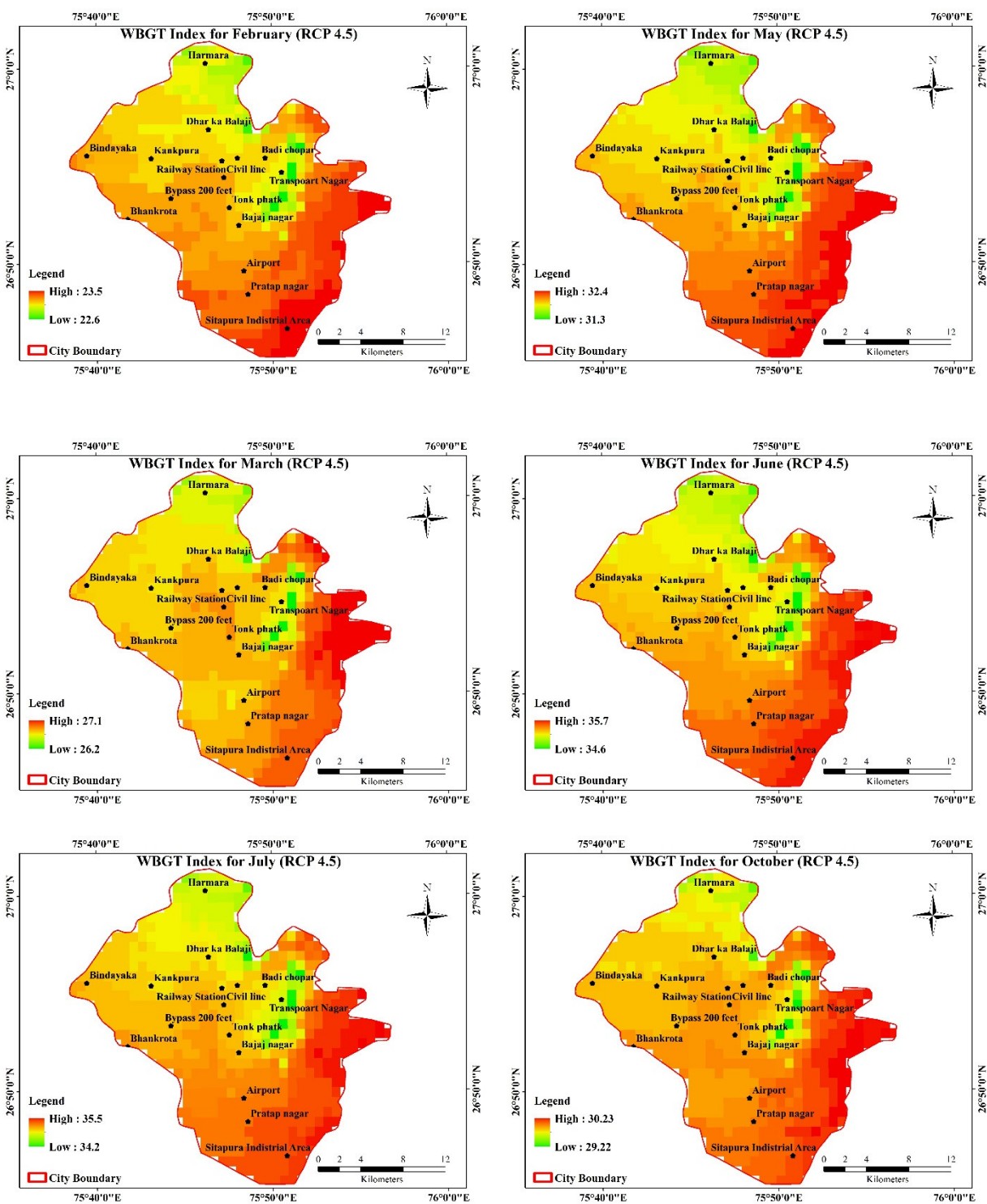

**Figure 10.** *Cont*.

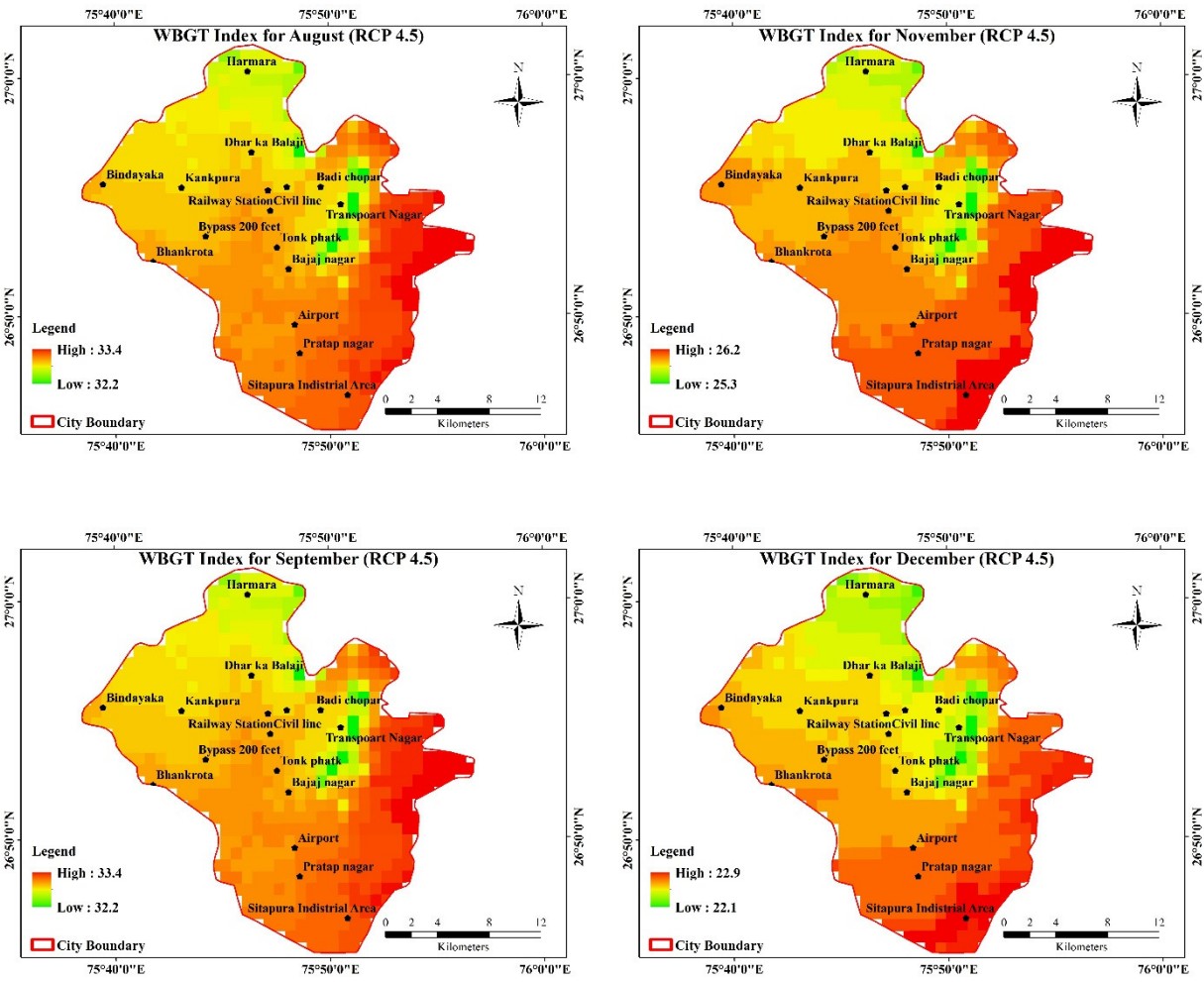

**Figure 10.** Spatial variation of WBGT for the future (RCP4.5) period.

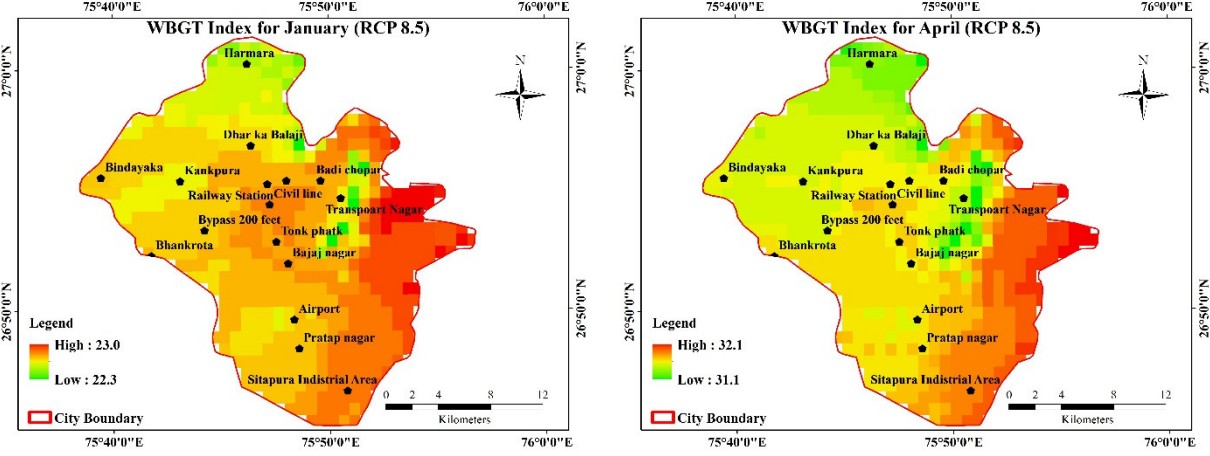

**Figure 11.** *Cont.*

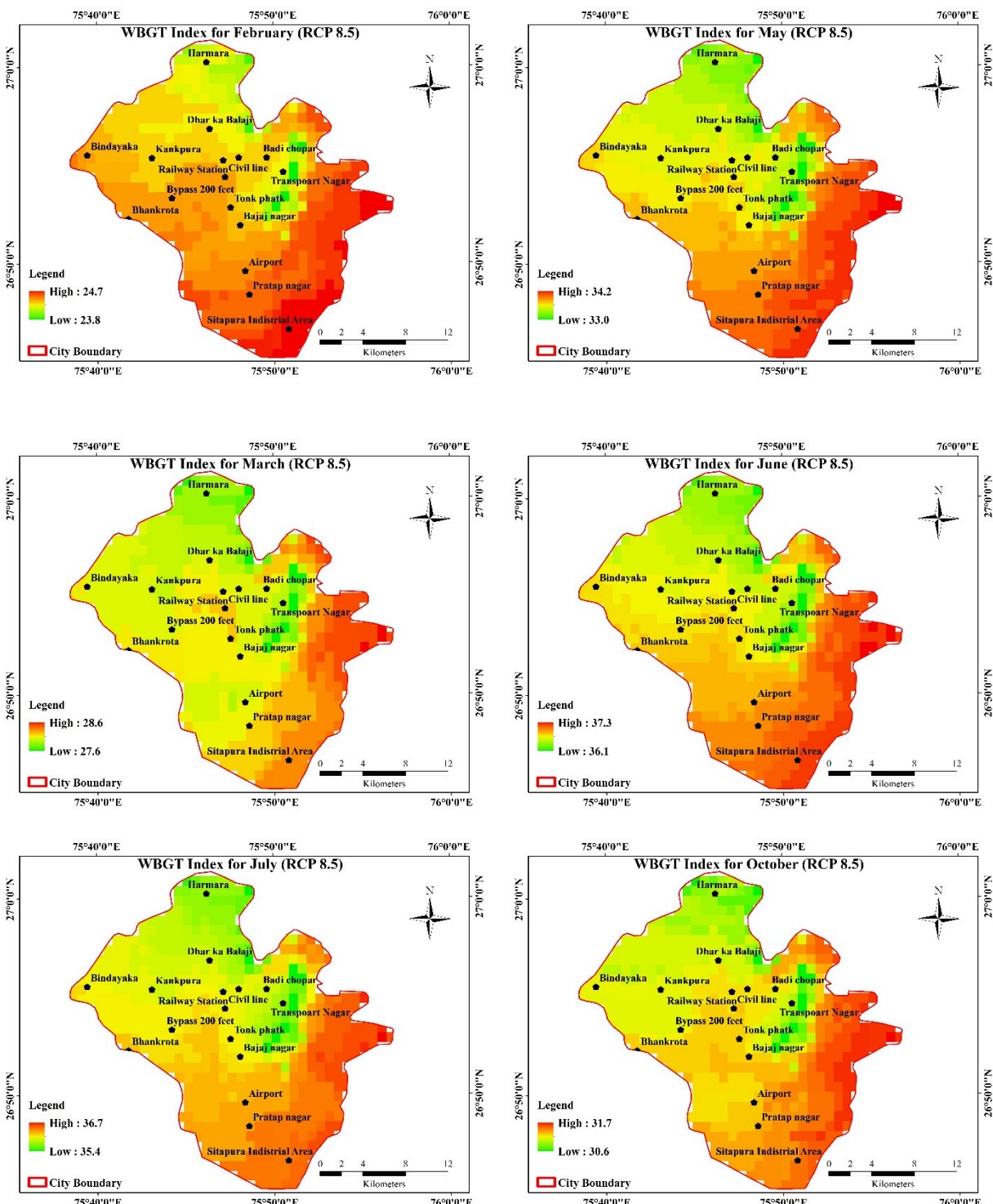

**Figure 11.** *Cont.*

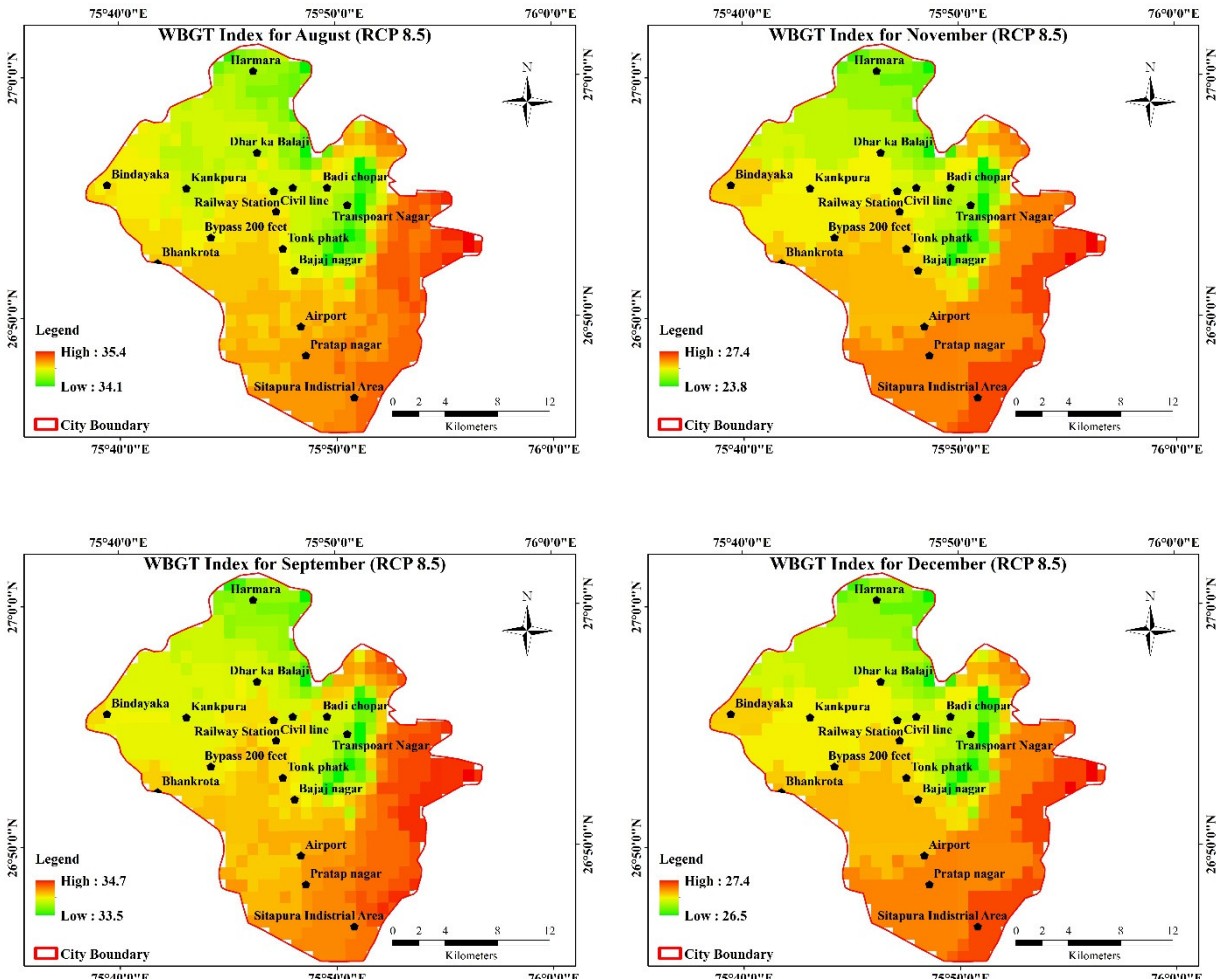

**Figure 11.** Spatial variation of WBGT for future (RCP8.5) period.

RCP8.5, which simulates a high emission scenario, consistently provides a large difference from RCP4.5. Figure 12 illustrates the disparity pattern, which is seen to be similar to the humidex indication. The study is being carried out to better understand how heat stress conditions vary seasonally. All four seasons' heat stress categories show little variation; however, the monsoon season shows a rise in the danger category. In the monsoon season, WBGT is at its highest; in the winter season, WBGT is at its lowest. An increasing value is found in the future period when compared to the historical period. Figure 13 shows the variation in monsoon WBGT for historical and future periods and the difference in indicator value within the city boundary. The WBGT is high in the summer and autumn season in the southeast direction and these changes are created due to the changes in land use pattern and expansion of urban areas.

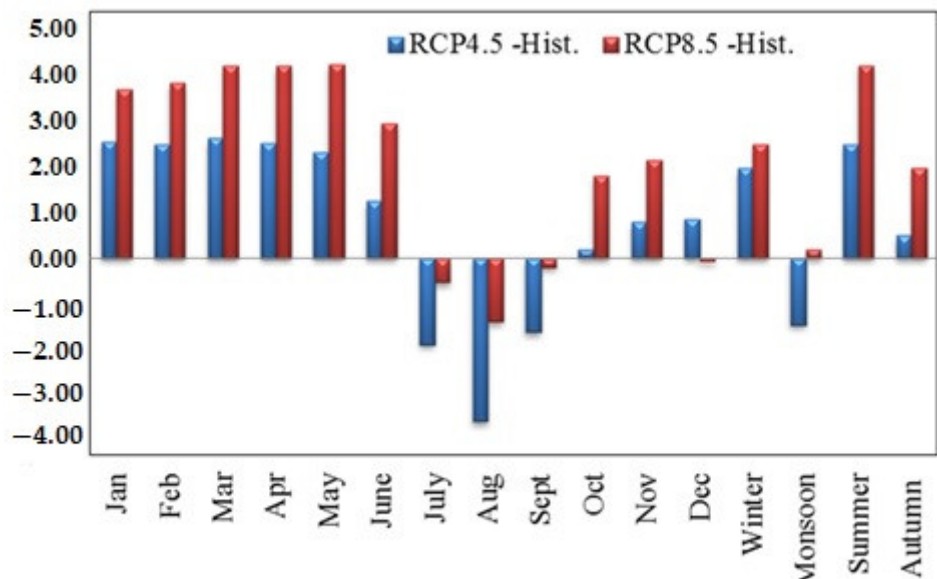

**Figure 12.** Difference in WBGT values for future scenarios (2050s).

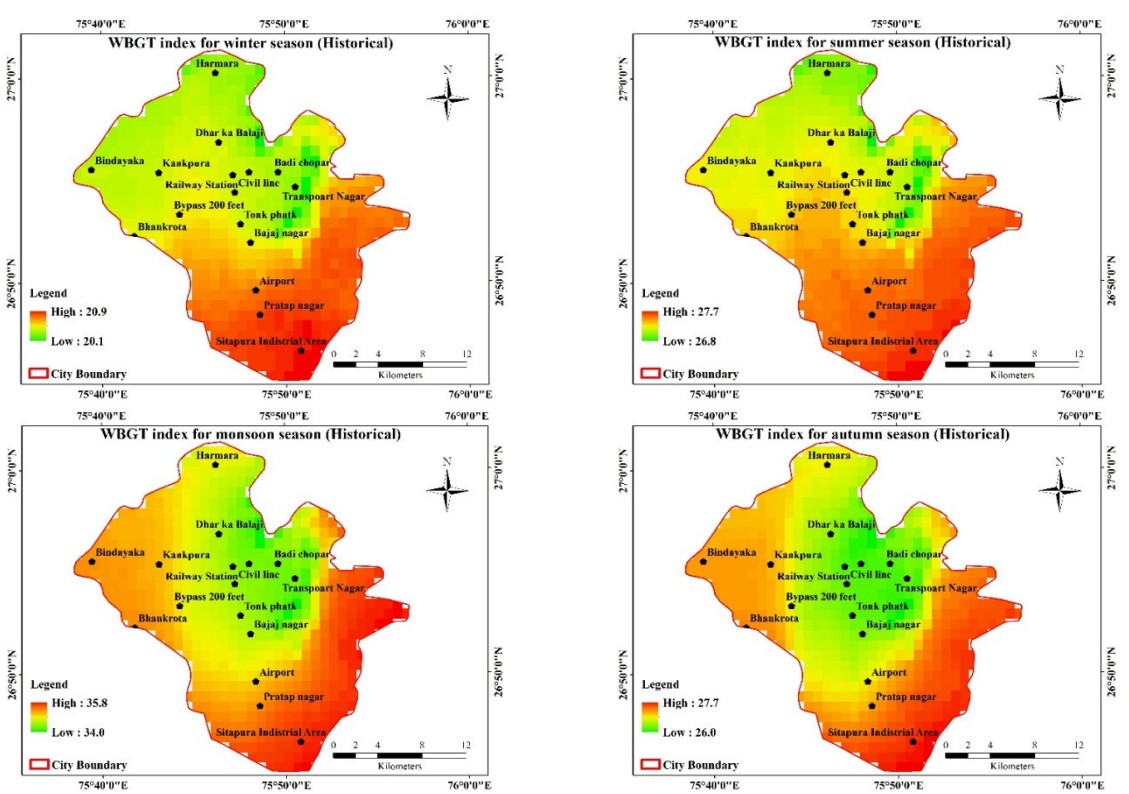

**Figure 13.** *Cont.*

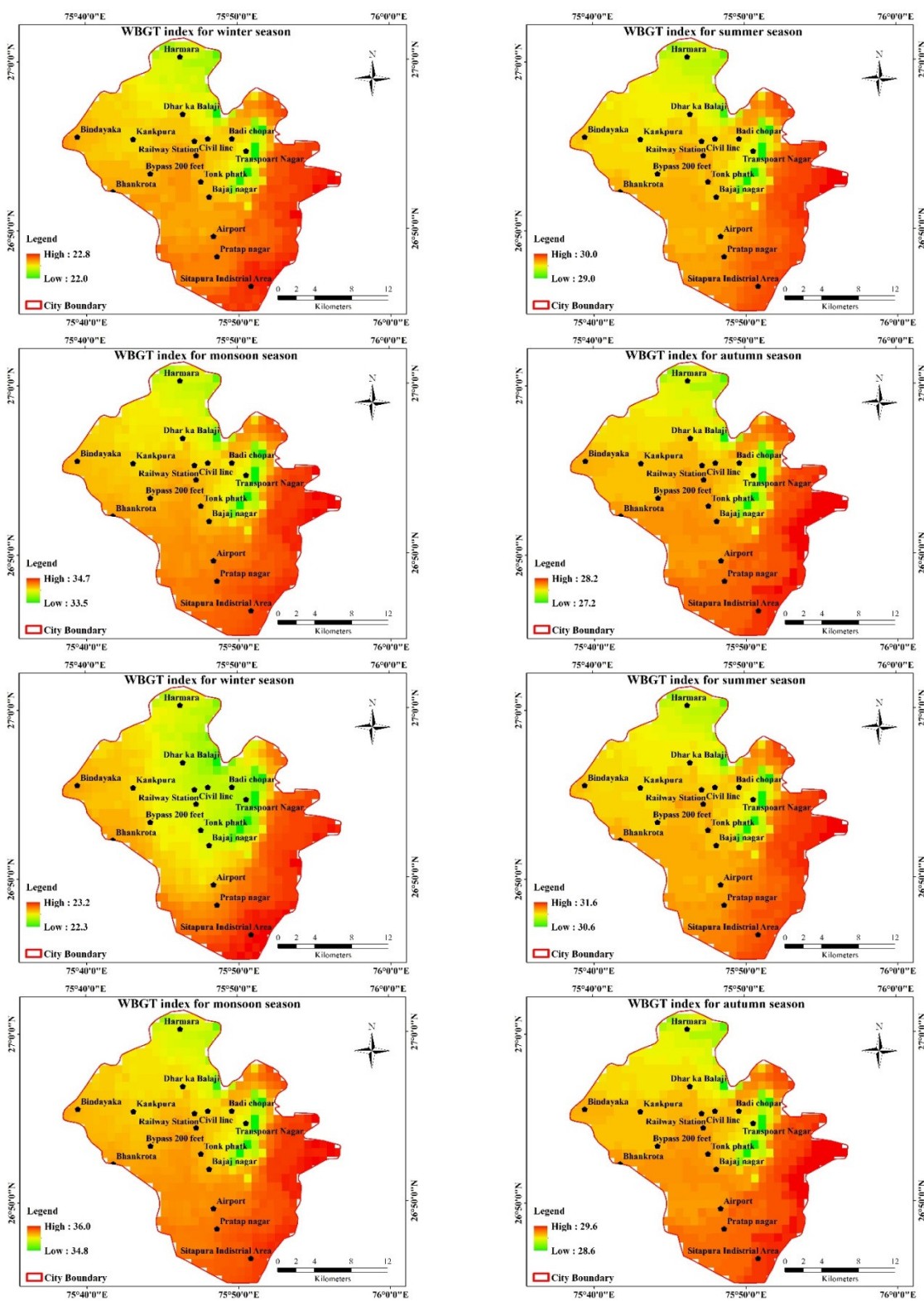

**Figure 13.** Seasonal map of the humidex index of the Jaipur city (Historical, RCP4.5, RCP8.5).

### 4.4. Normalized Difference Vegetation Index (NDVI)

In this study, NDVI was calculated for different periods: April 1993, April 2000, June 2010, and April 2015; NDVI values range from −1 to +1, different geographical features show the different NDVI values. These layers give different information through the bands and band 3 and 4 provides the vegetation with cover information of Jaipur city.

The extracted vegetation layer covers of NDVI were spatially compared with the colour composite image of Landsat-5 and Landsat-8 (TM and OLI) imagery. The range of NDVI in 1993 was −0.01 to 0.71, in 2000 was −0.019 to 0.63, and in 2010 was 0.04 to 0.56 of Landsat 5 TM imagery, and year 2015 shows the range of NDVI was −0.24 to 0.70 for the Landsat 8 OLI image of Jaipur city (Figure 14). The vegetation cover area utilizes solar radiation in the photosynthesis process and reduces the city's surrounding temperature.

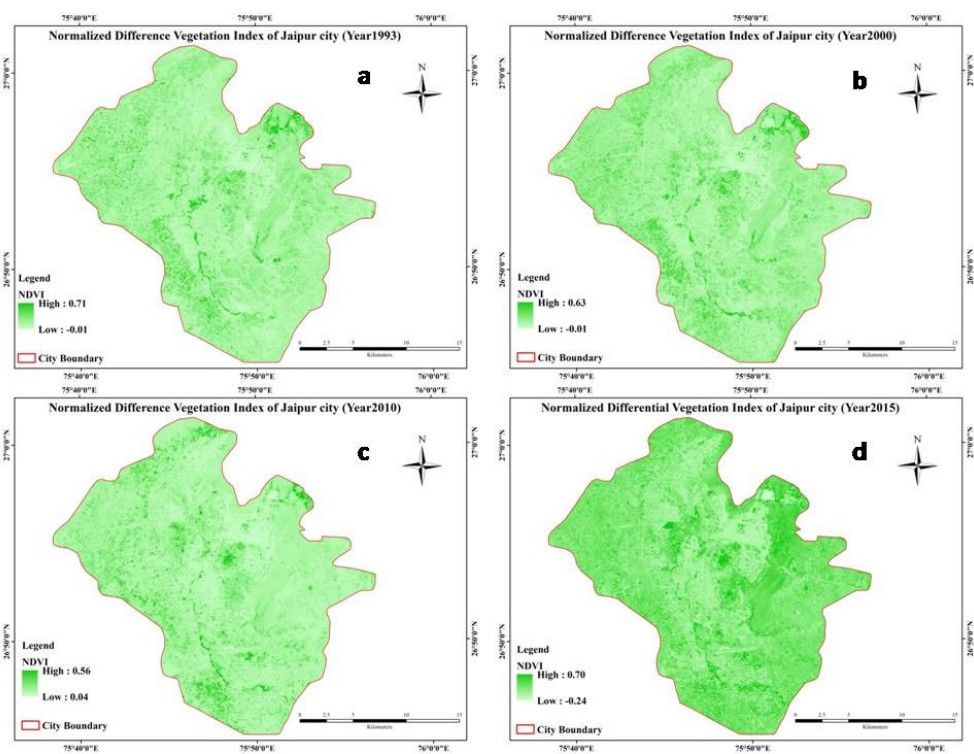

**Figure 14.** NDVI map of Jaipur city: (**a**) April 1993, (**b**) April 2000, (**c**) June 2010, and (**d**) April 2015.

*4.5. Soil-Adjusted Vegetation Index (SAVI)*

The study area is mainly classified into different types of land use. All these random samples are selected and the values of these cites are observed between NDVI and SAVI indices. The range of SAVI in 1993 was −0.005 to 0.50, in 2000 was −0.005 to 0.44, and in 2010 was −0.024 to 0.43 of Landsat 5 TM imagery, and 2015 shows the range of SAVI was 0.118 to 0.52 for the Landsat 8 OLI image of Jaipur city (Figure 15). This influence can be restricted using SAVI instead of NDVI. High NDVI and SAVI values were found in the buildup area.

On the other hand, there is a correlation between land use and NDVI data and data measured in meteorological stations in most research, including the current study, which is a significant reason for the efficiency of using this data for environmental issues. As a result, which can be derived indirectly using daily recorded metrological parameters in weather stations, it can be used to assess thermal conditions in Jaipur. Because evaluating environmental parameters for the calculation of heat stress indices is normally costly and time consuming, it is possible to alleviate this problem in environmental evaluations in open spaces by using daily recorded weather station data. Meteorological data has the advantage of being continuously recorded and providing a low-cost and comprehensive database for computing a variety of essential thermal indicators.

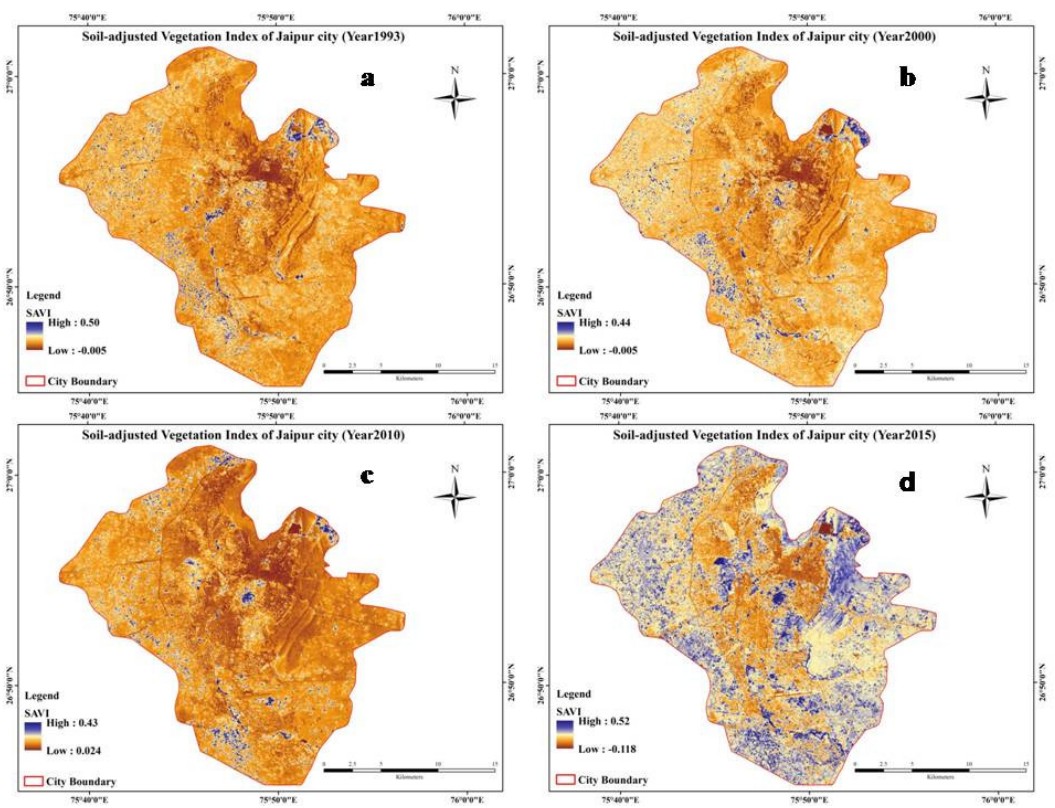

**Figure 15.** SAVI map of Jaipur city: (**a**) April 1993, (**b**) April 2000, (**c**) June 2010, and (**d**) April 2015.

## 5. Conclusions

The study's major goal was to predict the WBGT and humidex indexes for the past and future. This study demonstrates the expansion of urban land usage in Jaipur city from 1993 to 2015 using intermediate satellite images. The number of people living in cities has increased substantially in the last 23 years. The most prevalent type of land converted to urban areas is open terrain, followed by vegetation and hilly/rocky areas. The NDVI and SAVI indices are also used to determine the changes in land use patterns in the city and the amount of green space in the urban and peri-urban areas. The WBGT is highest during the monsoon season and lowest during the winter. When compared to the historical period, the future time shows an increase in value. Humidex's historical value has been 21.4, but it is projected to rise to 25.5 and 27.3 under the RCP4.5 and RCP8.5 scenarios. In May, the greatest humidex values were 39.5, 43.2, and 46.4 for the historical and two future RCP scenarios. The months of May and June are shown in the danger and extreme danger categories in the analysis. From January to May, the humidex values rise, then begin to fall until the month of December. It predicts that, with the exception of the monsoon season in the metropolis, discomfort levels will rise in the future. The findings indicate that the index's average value is increasing. Global warming or the absence of suitable conditions in these environments may be responsible for this trend. This will help in the identification of a better heat stress index for diverse situations and temperatures. There are some restrictions on the study; the distribution of indices studied throughout the different continents varies because the majority of studies undertaken in this field are focused on regions with hot climates. Application of WBGT and humidex indices has limited application in warmer climates as it shows a low level when the air temperature is in high range. The environmental heat index is preferred in occupational situations, but is not suitable at all work locations [33]. The possible reason for using these indices is comprehensiveness of the index for assessing the thermal stress conditions with limited data availability. However, it is a useful indices to understand the pattern of long-term

change and warning purposes. The additional limitations of this analysis were the dearth of pertinent papers and the evidence provided in the articles. Appropriate protective strategies are required to prepare for the working population, which includes vulnerable persons whose occupational health and performance are harmed by heat stress.

**Author Contributions:** Conceptualization: S.C. and D.S.; methodology: S.C., D.S. and S.K.D.; formal analysis: S.C., D.S. and S.K.D.; investigation: S.C. and D.S.; writing—original draft: S.C. and S.K.D.; writing—review and editing: D.S., B.K.M. and R.D. All authors have read and agreed to the published version of the manuscript.

**Funding:** This is supported by Central University of Rajasthan and the Strategic Research Fund of the Institute for Global Environmental Strategies.

**Institutional Review Board Statement:** Not applicable.

**Informed Consent Statement:** Not applicable.

**Data Availability Statement:** Not applicable.

**Conflicts of Interest:** The authors declare no conflict of interest.

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
