# Peer review of "Investigation of Spatio–Temporal Changes in Land Use and Heat Stress Indices over Jaipur City Using Geospatial Techniques"

_sustainability, doi:10.3390/su14159095_

Round 1
Reviewer 1 Report
First Impression
I believe the analyzing and finding are good. The authors try to combine different approaches to get the results. The main aim of the research is to investigate the spatio-temporal changes of Land Use and its impact on heat stress Indices over Jaipur City in India. The author also presented acceptable literatures for this work especially in the region. I believe that the results of the study are interesting and make a good contribution towards scientific society. I have some suggestions as follows,
1. Add more previous studies and compare what different between this study and other studies
2. More detail about studies areas
3. I suggest to have a flowchart for the methodology and data used for this study.
4. More details need in the section of methodology about the date were used in this studies
5. Authors need focus more in the results analysis and compare their results to any other similar research.
Reviewer 2 Report
This study is local, and the academic contribution is unclear. Unfortunately, it looks like a report, not a paper. I have doubts about whether a contribution like this one will be of interest to readers. The paper could perfectly be presented as a "handbook" rather than a research paper.
Line 42: UHI generally refers to the air temperature in the urban areas is generally higher than that in the rural areas. Please add the definition.
Line 59-90: Please cut down it. Please shorten the Introduction and focus on the summary of research gaps, objectives, or research questions.
The introduction section needs to modify with the latest references.
Please check the format of the references.
Study area: I want to know why the authors selected this study area, Add significance and importance of the study area in this section.
Line251 and other related lines: Change "figure" to "Figure".
The results and discussion section are in need of a large revision. The discussion session messed with the result. The discussion could be more in-depth, in particular with regard to the pons and cons of the methods to study the Urban Heat Stress/Heatwaves. Comments on sources of uncertainty and error are also suggested.
In conclusion, better to provide research limitations and future recommendations for other studies.
Reviewer 3 Report
The article Investigation of Spatio-temporal Changes of Land Use and Heat Stress Indices over Jaipur City using Geospatial Techniques presents the study of Urban Heat Stress / Heat using Bulb Globe Temperature (WBGT) and Humidex Index.
In the abstract use the expression or the years 1970-2000 and 2041-2060, these two indicators were measured in Jaipur. You can say that for the period 1970-2000 this indicator was measured but for the period 2041-2060 it was modeled / predicted.
Line 57 what is… (Hindustan times, 2017)?
To determine NDVi and SAVI, why did you choose June 2010? However, there is a big difference between April and June. Hence the cause of the big differences.
It would be useful at the end of section 4.5 to add some interdependence discussions between NDVi, SAVI and the WBGT and Humidex index.
The conclusions should be improved. What could be the potential stakeholders of this study?
What steps can be taken to eliminate the negative effects of Heat Stress Indices over Jaipur City?
The bibliography contains few works. There are countless scientific articles on this topic that should be mentioned and in the discussion section it would be useful to add the correlations between the results obtained by you and other studies in which the same methodology was used.
Round 2
Reviewer 2 Report
The authors did not address or did not fully address the following two comments.
UHI generally refers to the air temperature in the urban areas is generally higher than that in the rural areas. Please add the definition.
The results and discussion section are in need of a large revision. The discussion session messed with the result. The discussion could be more in-depth, in particular with regard to the pons and cons of the methods to study the Urban Heat Stress/Heatwaves. Comments on sources of uncertainty and error are also suggested.
Author Response
Response sheet – Reviewer 02
Point 1: UHI generally refers to the air temperature in the urban areas is generally higher than that in the rural areas. Please add the definition.
Response: Definition of UHI is added in the introduction section of the revised manuscript.
Point 2: The results and discussion section are in need of a large revision.
The discussion session messed with the result. The discussion could be more in-depth, in particular with regard to the pons and cons of the methods to study the Urban Heat Stress/Heatwaves. Comments on sources of uncertainty and error are also suggested.
Response: As suggested by reviewer, manuscript is revised in section Result and Discussion. Same is given in revised manuscript.
Reviewer 3 Report
The article has been improved! It can be published in this form. Congratulations to the authors!
Author Response
Comments and Suggestions for Authors The article has been improved! It can be published in this form. Congratulations to the authors! Response: Authors are thankful to reviewer for their comments and support.